# Bring *My* Cup! Personalizing Vision-Language-Action Models with Visual Attentive Prompting

**Sangoh Lee** [1]  **Sangwoo Mo** [2]  **Wook-Shin Han** [1]

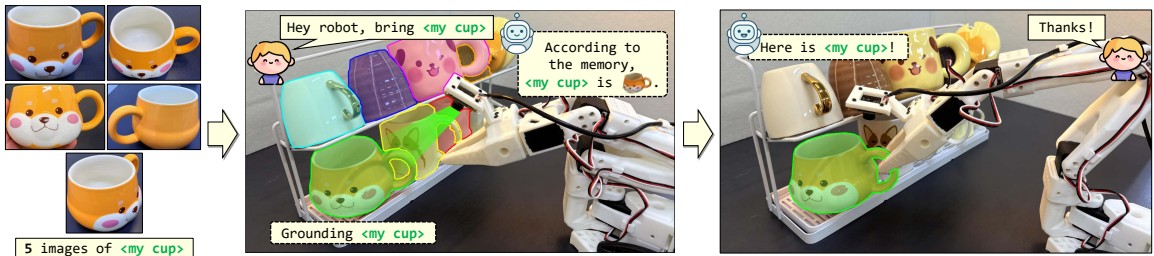

*Figure 1.* **Manipulating personal objects with VLA.** Existing vision-language-action (VLA) models cannot handle personal objects such as `<my cup>`, because they can only interpret generic, language-expressible semantics. We address this limitation with a simple framework, Visual Attentive Prompting (VAP). It first grounds the user-specific object in the scene by matching it against the memory and then uses visual prompting to guide the VLA. This pipeline enables existing VLA models to manipulate personal objects without any additional training.

## Abstract

While Vision-Language-Action (VLA) models generalize well to generic instructions, they struggle with personalized commands such as "bring *my* cup," where the robot must act on one specific instance among visually similar objects. We study this setting of manipulating personal objects, in which a VLA must identify and control a user-specific object unseen during training using only a few reference images. To address this challenge, we propose **Visual Attentive Prompting (VAP)**, a simple-yet-effective training-free perceptual adapter that equips frozen VLAs with top-down selective attention. VAP treats the reference images as a non-parametric visual memory, grounds the personal object in the scene through open-vocabulary detection and embedding-based matching, and then injects this grounding as a visual prompt by highlighting the object and rewriting the instruction. We construct two simulation benchmarks, Personalized-SIMPLER and Personalized-VLABench, and a real-world tabletop benchmark to evaluate personalized manipu-

lation across multiple robots and tasks. Experiments show that VAP consistently outperforms generic policies and token-learning baselines in both success rate and correct-object manipulation, helping to bridge the gap between semantic understanding and instance-level control.

## 1. Introduction

**"Hey robot, bring my cup."** You may ask for your coffee mug while getting ready for work, or have the robot fetch your pet's favorite toy from the living room floor. As illustrated in Figure 1, such scenarios where a user's personal belongings are mixed with visually similar objects will be a frequent necessity in daily life. To truly assist the user, the robot must discern the subtle visual details that distinguish a specific possession from other counterparts. However, general-purpose policies, such as Vision-Language-Action (VLA) models, typically fail to meet this requirement. Since user-specific instances and the personalized instructions used to refer to them (e.g., "my cup") are rarely covered during training, these models frequently collapse to category-level recognition and mis-ground the intended instance among same-category distractors.

Recent advancements in robot learning have empowered general-purpose policies to execute diverse manipulation tasks specified by natural language commands (Zitkovich et al., 2023; Kim et al., 2024; Intelligence et al., 2025). By training on large-scale robot datasets (Open X-Embodiment Collaboration et al., 2023), these models achieve strong generalization to generic instructions (e.g., "pick up the cup"). However, relying solely on language

---

[1]GSAI, POSTECH [2]IME, POSTECH. Correspondence to: Sangwoo Mo <sangwoo.mo@postech.ac.kr>, Wook-Shin Han <wshan@dblab.postech.ac.kr>.

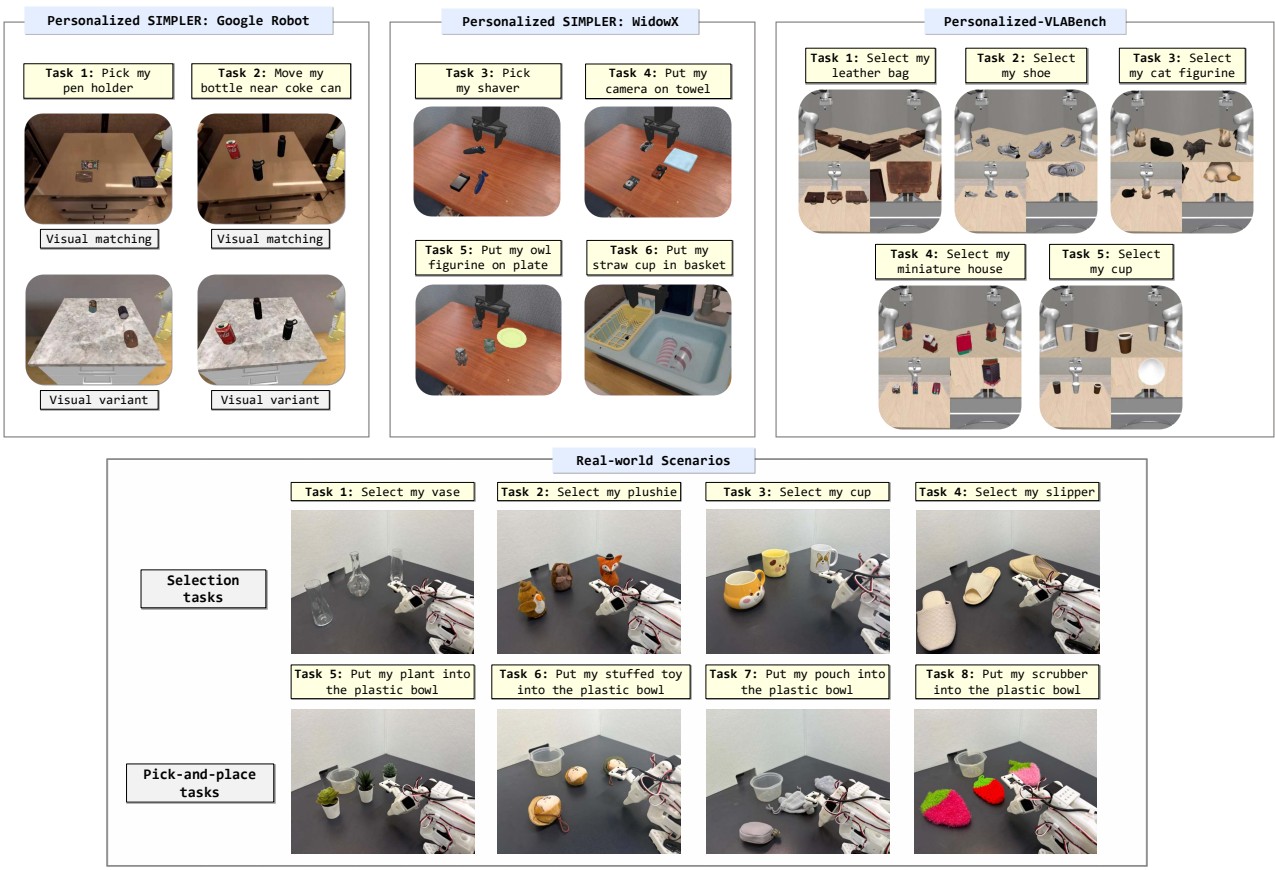

*Figure 2.* **Overview of Evaluation Benchmarks.** We evaluate in simulation (Personalized-SIMPLER, Personalized-VLABench) and on a real SO-101 robot. In each benchmark, one object is replaced by a user-specific instance, same-category distractors are added, and the policy must ground the correct instance from a few reference images.

creates a fundamental bottleneck for personalized assistance. Natural language inherently abstracts rich visual details into broad semantic categories. A standard VLA interprets a specific request for "my cup" simply as "a cup," effectively ignoring instance-level distinctions. This generalization leads the model to discard specific nuances, such as a unique pattern or a chip on the rim, that are critical for identifying the target among distractors. Even with detailed descriptions, text-only disambiguation remains brittle for VLAs because fine-grained instance cues must be grounded indirectly through language, as we demonstrate in Sec. 5. Moreover, per-object fine-tuning would require additional data and optimization for each personal item, which is impractical at deployment.

To bridge this semantic gap, we formulate the challenge of manipulating personal objects. Our goal is to adapt a frozen VLA to recognize and control a user's unique item unseen during training, utilizing only a handful of reference images. Specifically, given a small set of reference views, the agent must discern the specific target from visually similar distractors in a cluttered scene. This formulation targets instance-level generalization, extending VLA capabilities beyond broad category-level recognition.

We propose **Visual Attentive Prompting (VAP)**, a training-free framework that injects instance-awareness into frozen VLAs by intervening only on their inputs. Personalizing a VLA, however, is not simply a matter of providing a visual cue: in embodied manipulation the personal target must remain identifiable across every frame of closed-loop execution, even when the gripper or arm temporarily occludes it. VAP therefore couples (1) **Grounding**, which localizes the user's object using a small set of reference images, with (2) **Visual Prompting**, which overlays a mask-aligned highlight on the target and propagates this highlight across frames so that the policy attends to the same instance throughout the action sequence. The original instruction is rewritten in parallel to refer to the highlighted region, aligning visual and linguistic cues with a single instance signal. As a result, a new personal object can be used immediately after it is registered with a few reference images, without any retraining or fine-tuning.

We validate the efficacy of VAP across a comprehensive suite of simulation and real-world scenarios. Since standard VLA benchmarks primarily evaluate category-level generalization, we establish two new benchmarks, *Personalized-SIMPLER* and *Personalized-VLABench*, de-

signed to rigorously test instance-level identification among distractors (Figure 2, Top). Crucially, to demonstrate practical applicability, we extend this evaluation to a physical robot setup, tasking the agent with manipulating diverse personal objects in real-world (Figure 2, Bottom). Our experiments show that instance-level personalization for frozen VLAs can be achieved by an input-side adapter that *couples* reference-based grounding with an aligned visual highlight and instruction rewrite. Ablations confirm that neither component alone reliably closes the gap between semantic commands and instance-level control.

Our main contributions are as follows:

- **Personal Object Manipulation:** We introduce a personalization task for VLAs where the policy must manipulate user-specific objects among visually similar distractors using only a few reference images.
- **Visual Attentive Prompting (VAP):** We propose VAP, a simple-yet-effective input adapter that couples reference-based instance grounding with a mask-aligned highlight and a matched instruction rewrite, enabling frozen VLAs to act on user-specific instances.
- **Benchmarks and Evaluation:** We establish two simulation benchmarks and a real-world setup, demonstrating that VAP significantly outperforms generic and other baselines in personalized manipulation.

## 2. Related Work

**VLA Models.** VLA policies have emerged as a practical approach for general-purpose robotic manipulation (Kawaharazuka et al., 2025; Wen et al., 2025; Zitkovich et al., 2023; Kim et al., 2024; Black et al., 2024; Intelligence et al., 2025; Physical Intelligence et al., 2025). Trained on large-scale robot datasets (Open X-Embodiment Collaboration et al., 2023; Khazatsky et al., 2024; Walke et al., 2023; Fang et al., 2023), they can follow open-ended semantic commands and generalize to novel objects. However, recent benchmarks show that VLAs remain brittle under distribution shift and often struggle with fine-grained visual grounding (Fang et al., 2025; Zhang et al., 2025). This limitation is amplified in personalization, where the agent must identify a specific instance among lookalikes and act on previously unseen personal objects. We therefore study how to adapt frozen VLA backbones to personalized scenarios without modifying their parameters.

**Personalization in Foundation Models.** Personalization seeks to adapt large-scale priors to specific user contexts via tuning or conditioning. In LLMs, this involves lightweight adapters (Tan et al., 2024) or memory banks (Zhong et al., 2024). Similarly, text-to-image customization spans token optimization (Gal et al., 2023), weight tuning (Ruiz et al., 2023; Kumari et al., 2023), and encoder-based projection (Ye et al., 2023; Li et al., 2024),

extending even to segmentation (Zhang et al., 2024) and dialogue (Alaluf et al., 2024). However, these methods are predominantly confined to static generation or textual reasoning. They do not address the demands of embodied control, where an agent must not merely generate or describe a personal object, but physically interact with it in a dynamic environment.

**Personalization for Robotic Manipulation.** Prior personalization efforts have primarily targeted instruction modalities or high-level planning. For instance, VLAS (Zhao et al., 2025) incorporates speech recognition for voice commands, while TidyBot (Wu et al., 2023) and ProVox (Grannen et al., 2025) customize task scheduling. MEMENTO (Kwon et al., 2026) utilizes episodic memory to retrieve object semantics, and PIN (Barsellotti et al., 2024) conditions on reference images for navigation. However, these approaches do not address the *control* gap: enabling a general policy to physically manipulate a specific instance (e.g., "my cup") amongst visual distractors. We bridge this gap by constructing test-time visual prompts from reference images, allowing a VLA to reuse generic skills while manipulating the correct personal instance.

**Referring Object Manipulation.** A closely related line of work studies referring object manipulation, where the agent identifies and acts on a specific target indicated by an external referring signal in a cluttered scene. Prior work spans a broad range of language-based specifications: free-form referring expressions for grasping in clutter (Tziafas et al., 2023; Yang et al., 2022), category-attribute-spatial language with grounded priors (Huang et al., 2025; Vuong et al., 2024), interactive dialogue-based attribute disambiguation (Zhang et al., 2021; Yang et al., 2022), multi-modal prompts that include in-scene visual markers (Jiang et al., 2023), and image queries for general open-world manipulation (Stone et al., 2023). These approaches rely on language for disambiguation, with visual cues, when used, serving as scene-level markers rather than user-provided identity exemplars. Our setting instead specifies the target through reference photos of the same physical instance, the necessary signal when language cannot resolve visually similar candidates within the same category.

**Visual Prompting and Grounded Priors.** Visual prompting steers vision-language models by overlaying explicit cues, such as bounding boxes or masks, onto input images (Shtedritski et al., 2023; Yang et al., 2023). In robotics, recent works (Huang et al., 2025; Liu et al., 2024a; Nasiriany et al., 2024; Zheng et al., 2025) utilize these markers to inject spatial priors for improved control. In contrast, we leverage visual prompting for instance disambiguation. Instead of enhancing general manipulation capabilities, we use prompts to ground user-provided references, enabling a VLA to identify and act on specific

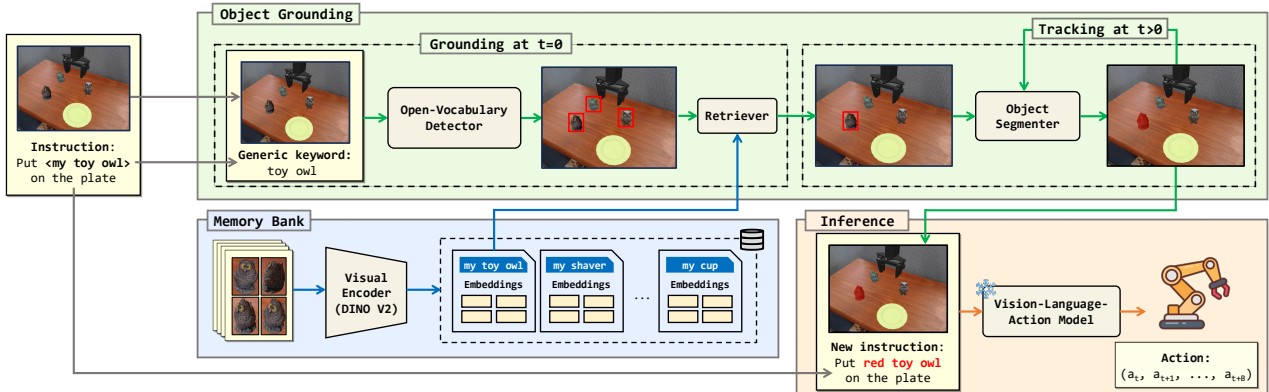

*Figure 3.* VAP builds a visual memory from a few reference images, grounds the target with frozen detection and segmentation, and prompts a frozen VLA by highlighting the object and rewriting the instruction. A tracker propagates the mask across frames, enabling training-free personalized manipulation.

personal objects among visually similar distractors. Static visual prompting alone is insufficient for embodied manipulation: per-frame re-grounding induces identity switches and cascading failures during closed-loop execution. VAP therefore couples visual prompting with spatio-temporal mask propagation, producing dynamically consistent identity prompts across the action sequence.

## 3. Visual Attentive Prompting

We propose Visual Attentive Prompting (VAP), a simple-yet-effective mechanism that steers VLA policies to manipulate user-specified instances without additional training. Unlike static visual–language personalization, embodied manipulation requires identity to remain consistent across every frame of closed-loop execution. VAP integrates grounding with spatio-temporal memory to maintain stable target identity throughout the action sequence.

### 3.1. Problem Formulation

We consider a pre-trained VLA policy $\pi_{\mathrm{VLA}}(a \mid x, \ell)$ mapping observation $x = (I, s)$ and instruction $\ell$ to action $a$, where $I = \{I^{(v)}\}_{v=1}^{V}$ denotes multi-view RGB images from $V$ cameras and $s$ represents the proprioceptive state. In our setting, the user specifies a target instance $o$ through two complementary signals: a small reference set $R_o = \{I_o^{(k)}\}_{k=1}^{K}$ (with $K \approx 5$) of photographs of the actual object, and an instruction containing a possessive reference to its coarse category (e.g., "my cup"). The category is known, but the specific instance is novel and unseen during training, and at test time the robot encounters $o$ amidst visually similar distractors of the same category. Reference images are the primary signal in this regime: detailed verbal descriptions cannot reliably distinguish two same-category instances, while a few photographs carry the discriminative visual cues (texture, decoration, subtle geometry) needed for instance-level identification. We there-

fore treat language as a category-level handle that the visual prompt later disambiguates.

Our goal is to construct prompted inputs $(\tilde{x}, \tilde{\ell})$ that guide the frozen $\pi_{\mathrm{VLA}}$ to manipulate the correct target using only $R_o$, without parameter updates. We formulate this not merely as prompting, but as searching for a visual intervention that maximizes policy success. While exact gradient-based optimization would be computationally prohibitive, we propose VAP as a zero-shot approximation. By explicitly overlaying the grounded mask with canonical visual attributes (e.g., solid colors), we bias the inputs toward simple attribute-language patterns (e.g., color adjectives) that are commonly grounded in pre-trained VLAs, while keeping the policy frozen.

### 3.2. Overview of VAP

VAP acts as a training-free inference framework that intervenes strictly in the input space. It first grounds the user's target instance in the scene using reference images, and then reshapes the visual and language inputs to provide an explicit pixel-level conditioning signal for the target region. This approach enables a generic manipulation model to identify and act on specific personal objects without any weight updates.

Formally, at timestep $t$, the robot receives its observation $x_t = (I_t, s_t)$, language instruction $\ell$, and a reference set $R_o$ depicting the user's object $o$. VAP uses these references to construct a prompted observation $\tilde{x}_t$ and instruction $\tilde{\ell}$ that direct the frozen VLA $\pi_{\mathrm{VLA}}$ toward the correct object. We formulate this process with a tracking-aware grounding function $g$ and a prompting function $p$:

$$M_t = g(I_t, R_o, \mathcal{H}_{t-1}), \qquad (\tilde{x}_t, \tilde{\ell}) = p(x_t, \ell, M_t)$$

where $g$ initializes $M_0$ from $(I_0, R_o)$ and propagates it for $t > 0$ using tracking history $\mathcal{H}_{t-1}$. The mask $M_t$ marks the pixels belonging to the personal object in the current

*Table 1.* Overview of our personalization benchmarks. We report the robot platform, task types, number of tasks and episodes, number of personal object categories, and camera views.

| Benchmark | Robot | Task types | #Tasks | #Episodes | #Personal categories | #Cameras |
|---|---|---|---|---|---|---|
| Personalized-SIMPLER (Fractal) | Google Robot | Pick / Move-near | 2 | 1685 | 2 | 1 |
| Personalized-SIMPLER (Bridge) | WidowX | Pick / Place | 4 | 96 | 4 | 1 |
| Personalized-VLABench | Franka | Selection | 5 | 250 | 5 | 3 |
| Real-world | SO-101 | Selection / Pick&Place | 8 | 160 | 8 | 3 |

images, and the prompting function $p$ turns this mask into the prompted inputs $(\tilde{x}_t, \tilde{\ell})$. The frozen VLA then acts on $(\tilde{x}_t, \tilde{\ell})$, so personalization is handled entirely by $g$ and $p$ operating on its inputs.

To ground user intent, VAP utilizes $R_o$ as a non-parametric visual memory. It matches the memory against the live images $I_t$ and selects the pixels that best match the user's object, producing a mask $M_t$ in each camera view. This process effectively maps an abstract reference (e.g., "my cup") into a concrete region in image space.

Once localized, VAP exposes the mask to the frozen policy through a coupled visual-and-language intervention. Visually, it overlays a semi-transparent highlight on the masked region while leaving the background context unchanged. In parallel, it rewrites the instruction so that the personal reference is replaced by a generic phrase matching the highlight (e.g., "pick up my cup" → "pick up the red object"). The two interventions are deliberately paired: the mask supplies a pixel-level anchor for where the target is, while the rewrite supplies the language token that tells the policy which anchor to attend to. Without either signal, the policy reverts to its category-level prior and selects among looka-likes essentially at random (Appendix E.1), so the mask and the rewrite act as a single binding mechanism rather than independent boosts.

### 3.3. Grounding and Prompting Details

We implement the grounding function $g$ via a coarse-to-fine perception pipeline: open-vocabulary detection proposes category-level candidates, reference matching identifies the target instance, and temporal tracking propagates the mask over time. In multi-view settings, we execute this pipeline independently for each camera view. We validate critical design choices—including instruction rewriting, voting aggregation, reference count, and prompt styles—via detailed ablations in Appendix E.

**Grounding.** We keep a memory $\mathcal{M}$ of user-specific objects. Offline, the user provides a small set of reference images for each personal item. In our benchmarks, we construct object-centric reference crops $R_o$ by detecting the target in each reference frame (with the same category-level detector used at test time) and cropping around the detected box to minimize background bias. A frozen visual encoder $f(\cdot)$ embeds each reference view $I_o^{(k)}$ of object $o$,

producing vectors $Z_o = \{f(I_o^{(k)})\}_{k=1}^K$.

At the start of an episode, we parse the personalized instruction (e.g., "bring my cup") and extract the generic category name $c$ ("cup"). We use $c$ as the text query for an open-vocabulary detector and run it on each camera view $I_t^{(v)}$, obtaining category-level bounding box proposals $B_t^{(v)} = \{b_i\}_{i=1}^{N_v}$. This stage isolates plausible category-level candidates, leaving specific instance identification to the subsequent matching step. If no candidates are detected ($N_v = 0$), we bypass prompting and pass the unmodified view to the VLA. We apply the same fallback if the tracker output mask is unavailable for a view.

Given the proposals $B_t^{(v)}$ and the reference embeddings $Z_o$, we retrieve the box that best matches the user's object. For each candidate $b_i \in B_t^{(v)}$, we crop $I_t^{(v)}[b_i]$, embed it with $f(\cdot)$ to obtain $e_i$, and compute cosine similarities $\cos(e_i, z_k)$ for all $z_k \in Z_o$. To aggregate evidence across reference views, we aggregate per-reference matches to select the target box:

$$b^* = \underset{b_i \in B_t^{(v)}}{\operatorname{argmax}} \sum_{k=1}^K \mathbb{1}\left[i = \underset{j}{\operatorname{argmax}} \cos(e_j, z_k)\right].$$

Each reference view $k$ votes for its most similar proposal, and we select the proposal with the largest vote count, favoring proposals supported consistently across references. If multiple proposals tie in vote count, we select among the tied proposals the one with the highest mean cosine similarity to the reference embeddings.

Once we choose $b^*$ for view $v$, we refine it into a pixel-wise mask $M_t^{(v)}$ using a class-agnostic segmenter, which yields the per-view masks required by $g$. For later timesteps ($t > 0$), we avoid rerunning detection and retrieval at every control step. Instead, we apply a real-time tracker that conditions on the current image $I_t^{(v)}$ together with its memory state $\mathcal{H}_{t-1}$, producing an updated mask $M_t^{(v)}$ and updating the memory online to follow the object as it moves.

**Visual Prompting.** The masks $\{M_t^{(v)}\}_{v=1}^V$ then instantiate the prompting function $p$. For each view $v$, we generate a highlighted image $\tilde{I}_t^{(v)}$ by overlaying a semi-transparent tint on $M_t^{(v)}$, using a mask-aligned cue to avoid highlighting irrelevant background. We leave the background unchanged, and we pass the proprioceptive state $s_t$ through unchanged. At the language level, we rewrite the origi-

nal instruction $\ell$ by replacing the personalized span (e.g., "my cup") with a generic phrase that matches the visual highlight (e.g., "the red cup"). Specifically, we apply a template-based rewrite by string matching "my X" and substituting it with "the tint-color X" (see Appendix A and E.1 for details). The frozen VLA is then evaluated on $(\{\tilde{I}_t^{(v)}\}_{v=1}^{V}, s_t, \tilde{\ell})$, which realizes the visual attentive prompting described in Section 3.2 while keeping the underlying policy training-free.

One might ask why we externalize the personal concept into a visual prompt rather than learning an internal token representation, as is standard in vision-language personalization. We find that such learned token embeddings do not transfer cleanly to embodied control: even when the token recognizes the target object in static images, the attention it induces in the frozen VLA drifts across frames during closed-loop execution and dissolves the instance signal exactly when the policy needs it (Appendix B). An external, mask-based intervention sidesteps this drift by re-anchoring the instance signal at every frame, which is what allows the policy's category-level skills to be reused on a specific personal instance without retraining.

## 4. Benchmarks

We evaluate VAP on simulation and real-world tabletop benchmarks that instantiate the personalization setting in Figure 2. Each benchmark requires a frozen VLA to act on a user-specific object instance given a few reference images, in the presence of same-category distractors. Table 1 summarizes benchmark statistics.

### 4.1. Simulation Benchmarks

To study personalization in simulation, we construct *Personalized-SIMPLER* and *Personalized-VLABench* by adapting SIMPLER (Li et al., 2025) and VLABench (Zhang et al., 2025). In both settings, we replace an object with a user-specific instance, add same-category distractors, and provide a small reference set $R_o$. Personalized-SIMPLER evaluates manipulation across two robots, whereas Personalized-VLABench isolates multi-view selection (Figure 2, top).

**Personalized-SIMPLER.** We personalize six tasks from SIMPLER (Li et al., 2025) across the *Fractal* (Google Robot) and *Bridge* (WidowX) settings. In each episode, a task-relevant object is replaced by a Sketchfab asset (Sketchfab, 2026) accompanied by distractors, using five reference images to form $R_o$. Following original evaluation protocols, we generate 1,685 Fractal episodes across visual-matching and variant-aggregation tracks (enumerating visual perturbations), and use the standard 96 episodes for Bridge (Table 1).

**Personalized-VLABench.** We adapt VLABench (Zhang et al., 2025) to five selection tasks on a Franka arm with three synchronized cameras. Focusing on categories like bags and shoes, each episode replaces one object with a user-specific instance (using Sketchfab assets and 5 reference images) amidst two same-category distractors. The benchmark comprises 250 episodes (5 tasks × 50 episodes), providing personalized instructions (e.g., "select my leather bag") and multi-view observations.

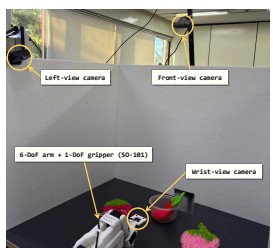 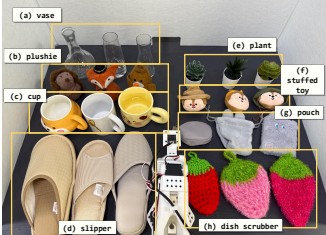

*(a)* The SO-101 tabletop setup with RGB cameras. *(b)* Real-world objects: one personal instance and two distractors.

*Figure 4.* Overview of our real-world experimental setup.

### 4.2. Real-world Benchmarks

We construct a real-world benchmark analogous to the simulation structure (Figure 2, bottom). The workspace features a 6-DoF SO-101 arm from the LeRobot platform (HuggingFace, 2026) and three RGB cameras (front, side, wrist). We target 8 everyday categories (e.g., cup, slipper), employing three unseen physical instances per category: one target and two lookalike distractors (Figure 4).

In each episode, we arrange the three instances and provide the system with the user's reference set $R_o$—captured and cropped directly from the physical cameras—alongside a personalized instruction (e.g., "select my slipper" or "put my scrubber into the bowl"). Spanning both selection and pick-and-place tasks, this benchmark rigorously evaluates whether VAP can reliably identify and manipulate user-specified objects on physical hardware.

## 5. Experiments

This section describes the experimental setup, metrics, and baselines used to evaluate VAP on the benchmarks in Section 4. Appendix A provides additional environment and hyperparameter details.

### 5.1. Experimental Setup

We evaluate VAP in both simulation and real-world settings to assess how well it personalizes a pre-trained VLA to user-specific objects.

**Backbone VLA.** Unless otherwise stated, we use $\pi_{0.5}$ (Intelligence et al., 2025) as the backbone VLA policy. For experiments on Personalized-SIMPLER, we instead use the $\pi_0$ checkpoint released for the Fractal and Bridge set-

*Table 2.* Performance on the **Personalized-SIMPLER** benchmark with the **Google Robot** platform. Track 1 evaluates visual matching under unseen personal objects, and Track 2 aggregates performance across systematic visual variants. VAP outperforms all baselines.

| Method | Task 1: Pick my pen holder | | Task 2: Move my bottle near coke can | |
|---|---|---|---|---|
| | CMR (%) | SR (%) | CMR (%) | SR (%) |
| *Track 1: Visual Matching with Unseen Personalization Objects* | | | | |
| $\pi_0$ | 10.5 | 8.5 | 50.4 | 48.6 |
| $\pi_0$ + Soft Prompt | 33.1 | 24.0 | 60.0 | 51.7 |
| $\pi_0$ + Hard Prompt (Short) | 11.0 | 8.9 | 66.7 | 57.5 |
| $\pi_0$ + Hard Prompt (Long) | 11.2 | 9.1 | 64.5 | 58.8 |
| $\pi_0$ + VAP | **89.2** | **60.3** | **83.2** | **75.0** |
| *Track 2: Variant Aggregation with Unseen Personalization Objects* | | | | |
| $\pi_0$ | 17.4 | 10.2 | 49.2 | 41.3 |
| $\pi_0$ + Soft Prompt | 31.8 | 26.4 | 57.7 | 44.7 |
| $\pi_0$ + Hard Prompt (Short) | 21.6 | 14.4 | 71.5 | 54.2 |
| $\pi_0$ + Hard Prompt (Long) | 17.3 | 12.6 | 67.7 | 51.3 |
| $\pi_0$ + VAP | **87.3** | **58.2** | **72.4** | **62.5** |

*Table 3.* Performance on the **Personalized-SIMPLER** benchmark using the **WidowX** platform across four manipulation tasks. For each task, we evaluate 10 runs of 24 episodes and report the mean. VAP's modular perception pipeline achieves consistently high success rates, whereas prior methods struggle to personalize.

| Method | Task 3: Pick shaver | | Task 4: Put camera | | Task 5: Put owl | | Task 6: Put straw cup | |
|---|---|---|---|---|---|---|---|---|
| | CMR (%) | SR (%) | CMR (%) | SR (%) | CMR (%) | SR (%) | CMR (%) | SR (%) |
| $\pi_0$ | 0.0 | 0.0 | 34.2 | 31.7 | 35.2 | 30.3 | 80.6 | 27.8 |
| $\pi_0$ + Soft Prompt | 25.0 | 8.3 | 54.2 | 41.7 | 79.2 | 62.5 | 87.5 | 29.2 |
| $\pi_0$ + Hard Prompt (Short) | 0.0 | 0.0 | 50.0 | 25.0 | 37.5 | 33.3 | 83.3 | 29.2 |
| $\pi_0$ + Hard Prompt (Long) | 0.0 | 0.0 | 53.2 | 27.9 | 36.0 | 31.6 | 83.4 | 30.1 |
| $\pi_0$ + VAP | **82.9** | **71.3** | **92.1** | **92.1** | **100.0** | **95.0** | **100.0** | **75.6** |

*Table 4.* Performance on the **Personalized-VLABench** benchmark. For each task, we evaluate 10 runs of 50 episodes and report the mean. VAP outperforms other baselines across all scenarios.

| Method | Task 1: Leather bag | Task 2: Shoe | Task 3: Cat figurine | Task 4: Miniature house | Task 5: Cup |
|---|---|---|---|---|---|
| $\pi_{0.5}$ | 33.6 | 30.4 | 35.2 | 29.6 | 40.4 |
| $\pi_{0.5}$ + Soft Prompt | 51.2 | 40.8 | 44.4 | 39.6 | 54.8 |
| $\pi_{0.5}$ + Hard Prompt (Short) | 52.4 | 37.6 | 44.0 | 38.8 | 51.6 |
| $\pi_{0.5}$ + Hard Prompt (Long) | 52.0 | 39.6 | 42.8 | 39.6 | 50.0 |
| $\pi_{0.5}$ + VAP | **89.2** | **54.0** | **51.6** | **52.4** | **60.8** |

tings in the open-source repository (Ren, 2026), following prior work showing that $\pi_0$ attains consistently strong performance on these settings (Fang et al., 2025). We fine-tune the base checkpoint $\pi_{0.5}$ solely for environment adaptation, using generic data that explicitly excludes personal objects and personalized instructions. The resulting backbone $\pi_{\text{VLA}}$ remains frozen throughout all experiments. Both VAP and the baselines share this identical policy to ensure a fair comparison. Appendix A.2 provides further details on this adaptation process.

**Evaluation Metrics.** We report Success Rate (SR), the fraction of episodes that complete the task, following standard VLA evaluations (Intelligence et al., 2025; Kim et al., 2024). For pick and pick-and-place tasks, we additionally report Correct Movement Ratio (CMR), the fraction of episodes in which the policy moves the target personal object at least once, regardless of final success (Sridhar et al., 2025). CMR is not defined for pointing or selection tasks, and higher is better for all metrics.

**Implementation.** For the perception module in VAP, we integrate state-of-the-art vision foundation models to en-

sure robust grounding: DINOv2 (Oquab et al., 2023) (final-layer $\ell_2$-normalized `[CLS]` token) for discriminative visual feature matching, Grounding DINO (Liu et al., 2024b) for open-vocabulary proposal, and SAM2 (Ravi et al., 2024) for consistent mask tracking across video frames.

### 5.2. Baselines

We compare VAP against the following baselines, all operating on the shared frozen backbone.

- **Generic VLA:** This baseline feeds the raw personalized instruction (e.g., "pick up my shaver") directly to the model. It serves as a lower bound, measuring the VLA's zero-shot ability to resolve personal references without any external grounding mechanism.

- **Hard Prompt (Short/Long):** A language-centric baseline that substitutes visual cues with text. We extract textual descriptions from the reference images and use an LLM to append these details to the instruction (generating Short or Long variants). This setup evaluates whether detailed linguistic context alone is sufficient for personalization (details in Appendix A.6).

- **Soft Prompt (Token Learning):** A token-learning baseline adapted from Yo'LLaVA (Nguyen et al., 2024). For each personal object, we optimize a specific token embedding within the VLA's language encoder using the reference images. During inference, we inject this learned token into the instruction sequence to represent the personal concept, while keeping the rest of the model weights frozen.

These three baselines isolate distinct hypotheses about whether personalization can be achieved without retraining: whether VLAs already handle personal references implicitly (Generic), whether sufficiently detailed language descriptions can disambiguate same-category instances (Hard Prompt), and whether learning concept tokens transfers to action prediction (Soft Prompt). To stress-test the last hypothesis in particular, we maximize the strength of the Soft Prompt baseline using Yo'LLaVA-style supervision and oracle hard negatives sampled from test-time objects. This optimized setup achieves > 95% accuracy on VQA recognition probes, and we further verify that the learned token transfers to the VLA with minimal embedding shift and successfully induces object-centric attention (Appendix B). Despite these favorable conditions, improvements remain marginal (Sections 5.3–5.4), indicating that surface-level interventions do not convert visual specificity into manipulation success.

### 5.3. Results on Simulation Benchmarks

We evaluate instance-level manipulation across three simulation benchmarks (Tables 2, 3, 4).

On **Personalized-SIMPLER** (Google Robot), VAP yields substantial gains in both correct-object interaction (CMR) and task completion (SR). The gap is most pronounced on the pen-holder task, where generic and language-based baselines frequently fail due to distractor interference. In the visual-matching track, VAP boosts SR/CMR from 8.5%/10.5% to 60.3%/89.2%, demonstrating that visual prompting effectively resolves instance ambiguity. Crucially, these gains persist under variant aggregation (SR 58.2%, CMR 87.3%), confirming robustness to visual perturbations. While Hard Prompts remain competitive on the simpler bottle task, VAP delivers the most consistent performance across all scenarios.

The **WidowX** (Bridge) results further support the robustness of VAP on manipulation tasks. In the picking task (Task 3), baselines struggle to reliably locate the target, whereas VAP achieves 71.3% SR. This advantage extends to placement tasks (Tasks 4–6), where VAP consistently outperforms all baselines, attaining > 90% SR on Tasks 4 and 5. While baselines occasionally register high interaction rates (CMR) in confined setups, they fail to translate this into successful task completion. Overall, VAP is the only method that consistently achieves high SR while maintaining high CMR across these manipulation tasks.

Finally, on **Personalized-VLABench** (Table 4), VAP demonstrates superior performance across all multi-view selection scenarios. While text-based prompts offer partial improvements by leveraging category-level semantics (e.g., distinguishing a bag from a cup), they fail to resolve instance-specific ambiguities. Consequently, VAP

*Table 5.* **Runtime analysis.** The initialization runs once per episode, while tracking and policy inference execute at every control step. VAP adds minimal overhead (0.02 s) to the control loop.

| Phase | Component | Time (s) |
|---|---|---|
| Initialization | Grounding DINO | 0.19 |
| | Segmentation & Embedding | 0.07 |
| Per Step | SAM2 Tracking | 0.02 |
| | VLA Policy Inference | 0.20 |

surpasses the strongest baseline by a significant margin (e.g., +36.8 points on the leather bag task), underscoring the necessity of explicit visual masks for robust personalization. We provide further analysis of the token-learning baseline and attention visualizations in Appendices B and B.3.

### 5.4. Results on Real-world Benchmark

We evaluate VAP on a real-world benchmark comprising four selection and four pick-and-place tasks (Figure 5). In selection tasks, text-based baselines struggle to distinguish personal objects from same-category distractors, plateauing at 40.0–45.0% SR. In contrast, VAP leverages explicit visual prompting to effectively disambiguate the target, boosting SR to 80.0% (vs. 30.0% for generic $\pi_{0.5}$).

This advantage extends to longer-horizon pick-and-place tasks. VAP improves average SR from 18.8% to 58.8%, significantly outperforming soft/hard prompts which remain in the 27.5–31.2% range. Crucially, baseline methods exhibit a large gap between initial interaction (CMR) and final success, often failing to complete the placement. VAP effectively closes this CMR–SR gap, demonstrating that explicit visual conditioning enables not only correct identification but also robust end-to-end execution in physical environments.

*Table 6.* **Failure analysis across benchmarks.** We report the overall failure rate for each setting. Cases 1–3 show the breakdown of specific error modes, calculated conditionally on failed episodes (— denotes not applicable).

| | P-SIMPLER (Fractal) | P-SIMPLER (Bridge) | Personalized-VLABench | Real-world Scenarios |
|---|---|---|---|---|
| **Fail (%)** | 37.6 | 16.6 | 38.4 | 31.9 |
| **Case 1** (%) | 13.1 | 21.4 | — | — |
| **Case 2** (%) | — | — | 59.2 | 50.9 |
| **Case 3** (%) | 86.9 | 78.6 | 40.8 | 49.1 |

### 5.5. Error Case Analysis

We categorize failure modes into three distinct types: single-view grounding errors (Case 1), multi-view inconsistency (Case 2), and control failures despite correct grounding (Case 3). Table 6 reveals distinct failure profiles depending on the setting. In single-view environments (Personalized-SIMPLER), failures are rarely caused by visual ambiguity (Case 1 < 22%) but are instead dominated

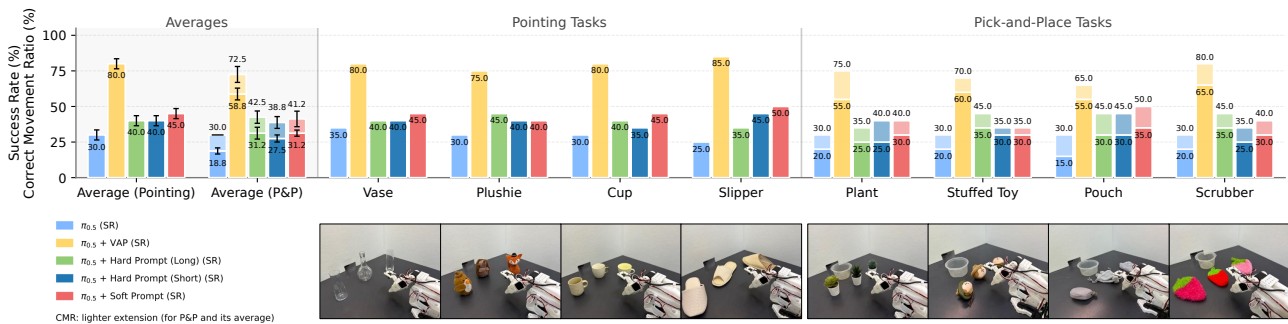

*Figure 5.* **Real-world performance.** We report SR over 20 trials for all tasks, and CMR specifically for pick-and-place tasks (inapplicable to selection). VAP achieves the highest SR/CMR across tasks. Error bars indicate the standard deviation across the four tasks.

by control limitations (Case 3). This indicates that while the visual prompt accurately localizes the target, the bottleneck lies in the VLA's physical manipulation capabilities.

Conversely, multi-view settings (Personalized-VLABench, Real-world) encounter a specific challenge: cross-view inconsistency (Case 2). Since VAP factorizes grounding and tracking per camera view for real-time inference, occlusions or partial visibility can cause per-view matching/tracking to lock onto different same-category instances across cameras. This yields contradictory mask prompts across views, which can destabilize behavior or bias actions toward the wrong object. A natural extension is to enforce explicit cross-view association/evidence fusion so that all views agree on a single target identity before prompting. We provide qualitative examples and several plausible design directions in Appendix D (see also Appendix D.2), and leave their implementation to future work.

Two architectural observations follow from this failure profile. First, the sequential factorization of grounding and manipulation does not itself bound performance: reliable spatio-temporal tracking maintains target identity through several seconds of complete invisibility, so grounding errors do not cascade into manipulation failures (Appendix E.5). Second, although perception accounts for only a minority of failures (Case 1, Table 6), VAP's modular factorization makes any remaining grounding error explicitly diagnosable (e.g., via the $N_v=0$ rejection condition) rather than silent. This is also a forward-looking property: as open-vocabulary detection and segmentation models continue to improve, VAP inherits those gains directly, without retraining or design changes on the policy side.

### 5.6. Efficiency of VAP

We assess the computational overhead of our perception modules in Table 5. VAP introduces a negligible one-time initialization cost (0.26 s) to establish the personalized mask using Grounding DINO and SAM2. During online control, the computational bottleneck remains the VLA policy inference itself (0.20 s/step). Critically, the SAM2-based mask tracking adds only 0.02 s per frame, incurring

a minimal 10% overhead to the control loop. This confirms that VAP enhances performance without compromising the real-time responsiveness of the robotic system.

### 5.7. Comparison to Visual Prompting Alternatives

A natural follow-up question is whether VAP's specific choice of mask-aligned identity tinting is responsible for the gains, or whether any visual prompt would suffice. Within the same frozen VLA, we additionally compare against a broad set of visual prompting strategies adopted in prior robotics work, including opaque masks, bounding boxes, points, circles, trajectory arrows, and numbered markers. On the harder WidowX setting, mask-aligned identity tinting reaches 83.5% SR, while points, trajectory arrows, and numbered markers stay between 28 and 36% (Appendix E.4). The gap reflects a structural distinction in what is being specified: spatial- and affordance-style prompts answer "where to act" on a per-step basis, whereas personalization requires "which instance" to remain stable across the action sequence. Mask-aligned tinting binds identity to pixel coverage rather than to a spatial pointer, which is what closes the gap.

## 6. Conclusion

In this work, we formalize the task of personalized object manipulation for VLAs, where the policy must distinguish and manipulate specific instances among visually similar distractors. To address this challenge, we propose Visual Attentive Prompting (VAP), demonstrating that an explicit visual prompting strategy can personalize frozen VLA models at inference time without any per-object parameter updates. Across diverse simulation and real-world benchmarks, our lightweight approach significantly outperforms language-based baselines in both object disambiguation and manipulation success. While promising, the current framework remains contingent on the performance of the underlying segmentation models and the consistency of masks across multi-view setups. Future work will focus on extending this paradigm to handle more complex personalization scenarios, such as user-specific spatial preferences and multi-step long-horizon tasks.

# Acknowledgements

This work was partly supported by Institute of Information & communications Technology Planning & Evaluation (IITP) grants funded by the Korea government (MSIT) (No. RS-2024-00509258, Global AI Frontier Lab, 50%; No. RS-2026-25524173, Ultra-Long-Term Hierarchical Memory and Reasoning Architecture for Next-Generation Omnimodal Agents, 25%), and the National Research Foundation of Korea (NRF) grant funded by the Korea government (MSIT) (No. RS-2025-00517736, 25%).

# Impact Statement

This paper presents work whose goal is to advance the field of machine learning. There are many potential societal consequences of our work, none of which we feel must be specifically highlighted here.

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

# A. Experimental Details

## A.1. Hardware and Software Settings

All experiments were conducted on a Linux server equipped with an Intel Xeon Gold 6230 CPU @ 2.10 GHz, 1 TB of RAM, and four NVIDIA RTX A6000 GPUs, running Ubuntu 22.04.3 LTS. Unless otherwise specified, VLA inference used a single RTX A6000 GPU. For training the VLA policies used in our experiments, we used four NVIDIA A100 GPUs. Our code is implemented in Python using PyTorch and builds on the public $\pi_0/\pi_{0.5}$ implementation.

## A.2. Training VLAs

We use the released checkpoint of $\pi_0$ for Personalized-SIMPLER, and train $\pi_{0.5}$ for Personalized-VLABench and the real-world experiments. This subsection describes the dataset construction and training configurations for $\pi_{0.5}$.

**Training Dataset for Personalized-VLABench.** We base our dataset on the `select_painting` task in VLABench. The original task requires the agent to press a red button corresponding to a target painting, which introduces an extra gap between object localization and motor control. To better match our personalized object manipulation setting, we modify the task so that the agent directly touches the target object on the table instead of pressing a button.

For each episode, we spawn a single painting at a random position on the table and issue the generic instruction *"select the painting"*. We use the RRT-based motion planner provided by VLABench to generate expert trajectories. Rather than directly copying the planner states, we execute the planned waypoints on the robot via inverse kinematics and record the resulting joint states. Given a state sequence $\{s_t\}_{t=0}^T$, we train the policy to predict state deltas $a_t = s_{t+1} - s_t$ instead of absolute states, which we found to yield more stable behavior in downstream control. We generate 500 training episodes with randomized object positions. Figure 6 shows example episodes from the Personalized-VLABench training data.

**Training Dataset for Real-world Scenarios.** To adapt $\pi_{0.5}$ to the real-world tabletop setting, we collect a small, task-specific dataset using teleoperation on the SO-101 arm. We define five tasks: two selection tasks (*"point at the cup"*, *"point at the cat figurine"*) and three pick-and-place tasks (*"put the miniature house into the plastic bowl"*, *"put the owl figurine into the plastic bowl"*, *"put the toy bus into the plastic bowl"*). All training objects in this dataset are disjoint from the personal objects used at evaluation time, so the policy only learns generic selection and pick-and-place skills rather than memorizing specific items.

For each task, we collect 50 teleoperated episodes at 30 Hz using the onboard RGB cameras, resulting in 250 episodes in total. As in the simulation setting, we train the policy to predict joint-space deltas between consecutive states. This dataset allows $\pi_{0.5}$ to adapt to the kinematics, camera viewpoints, and workspace of the real-world setup while keeping the personal objects unseen during training. Figure 7 and 8 show example episodes from the real-world scenario training data.

## A.3. Instruction Rewriting Logic

As described in Section 3.3, VAP employs instruction rewriting to align commands with visual prompts. We align our design with prior personalization frameworks, such as DreamBooth (Ruiz et al., 2023) and Yo'LLaVA (Nguyen et al., 2024), which typically rely on specific learned tokens or identifiers to explicitly trigger personalized concepts within user commands. Following this convention, we treat the specific possessive span (e.g., *"my cup"*) as the designated trigger for our personalization module. However, our framework is not limited to rigid templates and can be naturally extended to handle diverse natural language variations via semantic parsing (e.g., using LLMs or embedding similarity).

In this work, however, we adopt a deterministic rule-based approach to isolate the effect of visual prompting. The process proceeds as follows: We first parse the object category $c$ (e.g., *"cup"*) from the designated trigger *"my $c$"*. Then, we map the visual mask's fixed color to its text equivalent $w_{color}$ (e.g., *"red"*) and mechanically replace the trigger with *"the $w_{color}$ $c$"*. This explicit mapping effectively bridges the user's personalized intent with the visual cue provided to the VLA. Although we use this deterministic parser for the main results, the front-end is swappable: it can be replaced with a more capable semantic parser without modifying the grounding pipeline, which we empirically verify in Table 7 below.

**Training settings.** For both Personalized-VLABench and the real-world dataset, we initialize $\pi_{0.5}$ from the public checkpoint and fine-tune it with the same hyperparameters, except for the action horizon. We use a batch size of 64, an action dimension of 32, and train for 60,000 gradient steps using Adam. We set the action horizon to 4 for Personalized-VLABench and 32 for the real-world dataset. We train on a single node with four A100 GPUs and use the final iteration as the checkpoint. Our environment-adaptation schedule (hundreds of episodes and tens of thousands of gradient steps) follows common practice in prior VLA fine-tuning on new environments.

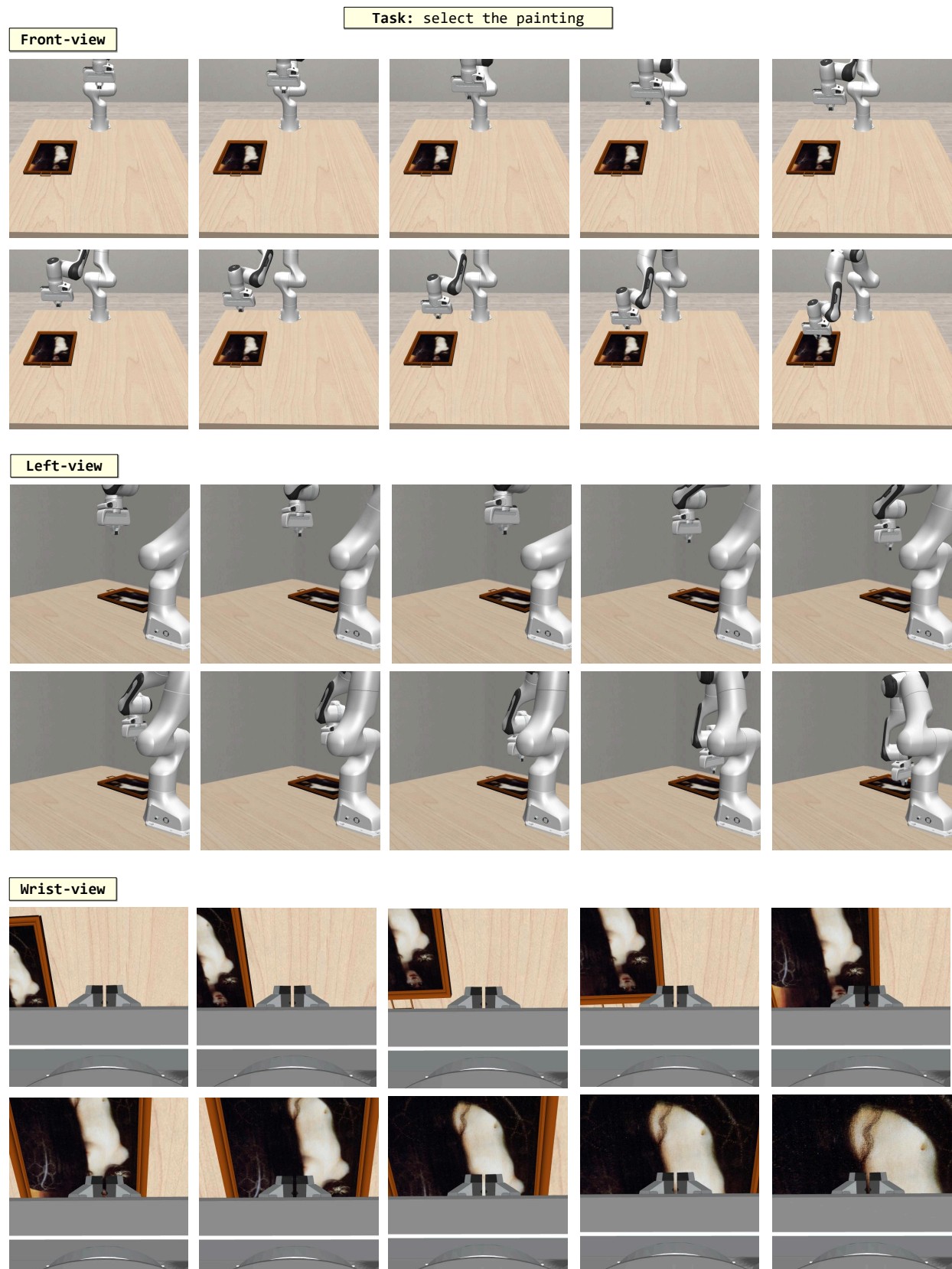

*Figure 6.* Example training sample for Personalized-VLABench

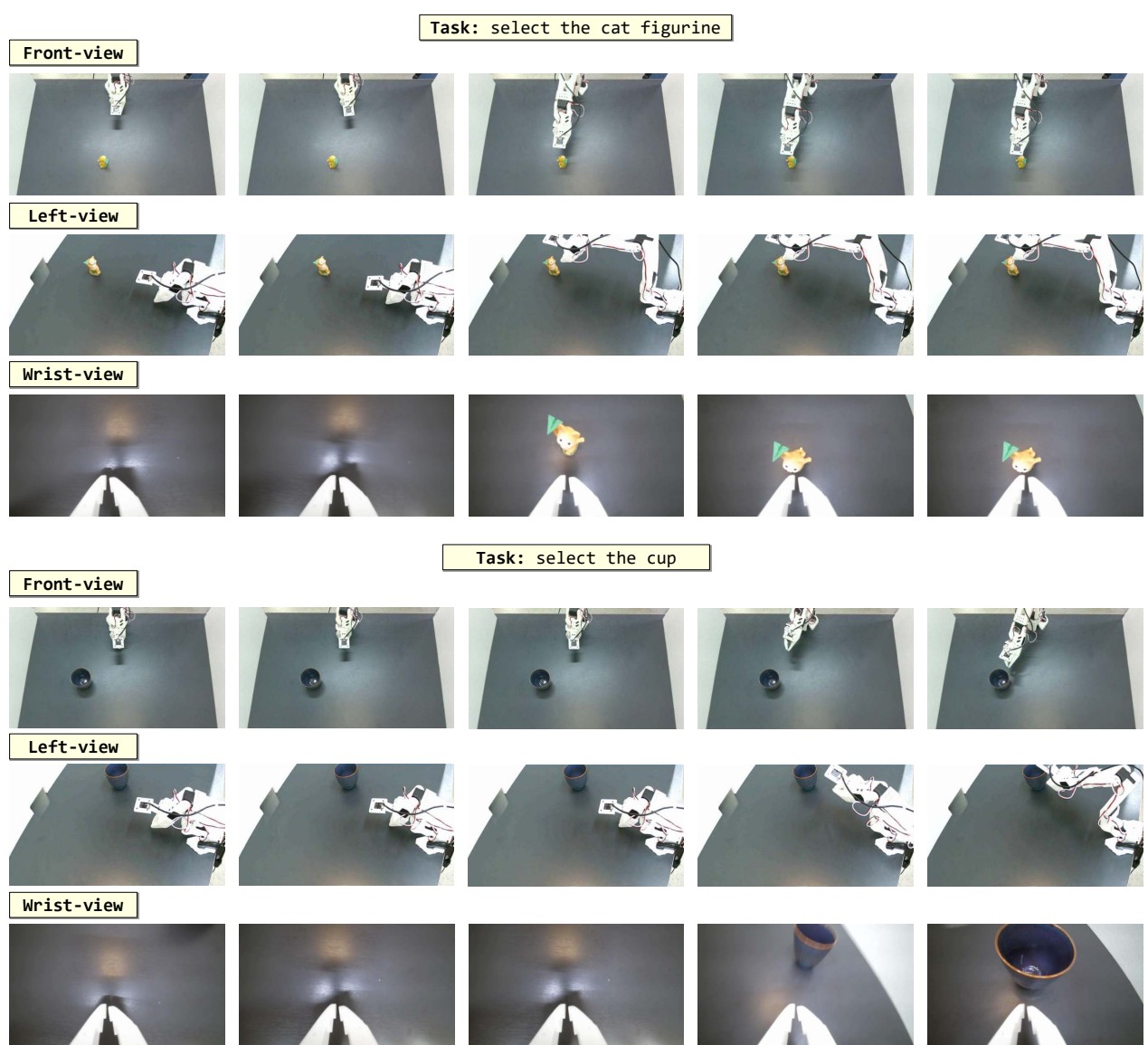

*Figure 7.* Example training sample for real-world scenarios (selection)

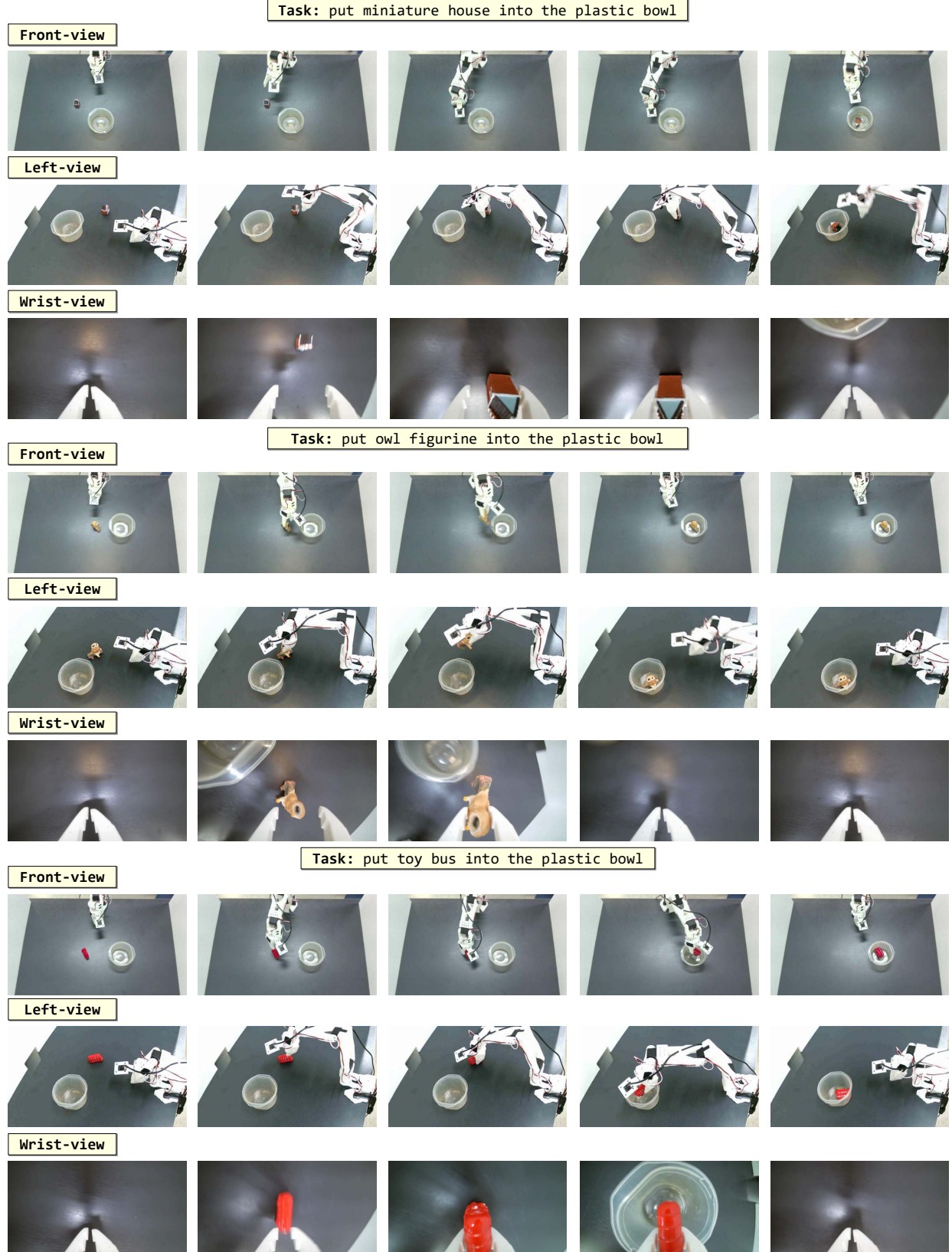

*Figure 8.* Example training sample for real-world scenarios (pick-and-place)

**Verifying Parser Swappability with a Lightweight LLM.** To confirm that the rule-based parser is a swappable front-end rather than a structural limitation, we replace it with a lightweight LLM (GPT-4o-mini) that resolves implicit references (e.g., *"the one I used earlier"*) by attending to the memory bank context. We evaluate parsing accuracy (PA, whether the LLM identifies the intended object) and task success rate (SR) on Personalized-SIMPLER. As shown in Table 7, the LLM-based parser resolves implicit expressions with PA above 94% across all tasks, and the resulting SR is comparable to the explicit-template setting, indicating that VAP integrates with richer front-end reasoning without modifications to the core grounding pipeline.

*Table 7.* Replacing the rule-based parser with a lightweight LLM (GPT-4o-mini) for implicit referring expressions. Implicit instructions are paraphrased to remove explicit *"my X"* templates (e.g., *"the one I used earlier"*).

| Task | Explicit, SR (%) | Implicit, PA (%) | Implicit, SR (%) |
|---|---|---|---|
| T1: Pick pen holder (Fractal) | 60.3 | 94.7 | 59.5 |
| T2: Move bottle (Fractal) | 75.0 | 96.2 | 72.0 |
| T3: Pick shaver (Bridge) | 71.3 | 98.4 | 71.6 |
| T4: Put camera (Bridge) | 92.1 | 97.8 | 91.3 |
| T5: Put owl figurine (Bridge) | 95.0 | 96.0 | 91.2 |
| T6: Put straw cup (Bridge) | 75.6 | 99.6 | 75.9 |

## A.4. Benchmark Construction Details

We summarize the construction and evaluation protocol of all four benchmarks. Table 8 reports per-benchmark metadata, and Table 10 provides a per-object task specification.

*Table 8.* Benchmark overview. "2 same-cat." indicates two same-category lookalike distractors.

| Benchmark | Object Categories | Object Source | Distractors | $K$ | Ref Source | Hz | Eval Protocol |
|---|---|---|---|---|---|---|---|
| P-SIMPLER (Fractal) | pen holder, bottle | Sketchfab 3D | 2 same-cat. | 5 | Rendered | 3 | Track 1: visual matching (460 ep) + Track 2: variant aggregation (1225 ep) |
| P-SIMPLER (Bridge) | shaver, camera, owl figurine, straw cup | Sketchfab 3D | 2 same-cat. | 5 | Rendered | 5 | 10 runs × 24 episodes |
| P-VLABench | leather bag, shoe, cat figurine, miniature house, cup | Sketchfab 3D | 2 same-cat. | 5 | Rendered | 10 | 5 tasks × 50 episodes |
| Real-world | vase, plushie, cup, slipper, plant, stuffed toy, pouch, scrubber | Offline retail | 2 same-cat. | 5 | Cropped from RGB | 30 | 20 trials/task |

## A.5. Train/Test Object Splits

All evaluations use disjoint training and test objects so that VAP's improvements reflect instance-level identity binding rather than backbone memorization. Table 9 summarizes the splits across the four benchmarks.

*Table 9.* Training and test object splits. P-SIMPLER uses public $\pi_0$ checkpoints with no further fine-tuning, so all evaluated objects are unseen personal instances. P-VLABench and Real-world use $\pi_{0.5}$ fine-tuned on disjoint generic objects.

| Benchmark | Training objects | Test objects | Train–test relation |
|---|---|---|---|
| P-SIMPLER (Fractal) | — | pen holder, bottle | unseen personal instances |
| P-SIMPLER (Bridge) | — | shaver, camera, owl figurine, straw cup | unseen personal instances |
| P-VLABench | painting | leather bag, shoe, cat figurine, miniature house, cup | unseen object categories |
| Real-world | cup, cat figurine, miniature house, owl figurine, toy bus | vase, plushie, cup, slipper, plant, stuffed toy, pouch, scrubber | unseen object categories (except cup) |

*Table 10.* Per-object task specifications: task type, maximum control horizon (#Steps), and intrinsic object property.

| Benchmark | Category | Task | # Steps | Object Property |
|---|---|---|---|---|
| Fractal | pen holder | Pick | 80 | Rigid, mesh texture |
| | bottle | Move-near | 80 | Rigid, glossy, red cap |
| Bridge | shaver | Pick | 60 | Rigid, complex geometry |
| | camera | Pick & Place | 60 | Rigid, small, glossy |
| | owl figurine | Pick & Place | 60 | Rigid, textured ornament |
| | straw cup | Pick & Place | 120 | Rigid, striped pattern |
| VLABench | leather bag | Selection | 150 | Deformable, leather |
| | shoe | Selection | 150 | Deformable, fabric/rubber |
| | cat figurine | Selection | 150 | Rigid, plush-like |
| | miniature house | Selection | 150 | Rigid, multicolor |
| | cup | Selection | 200 | Rigid, plastic |
| Real-world | vase | Selection | 150 | Rigid, transparent glass |
| | plushie | Selection | 150 | Deformable, soft fabric |
| | cup | Selection | 150 | Rigid, ceramic, glossy |
| | slipper | Selection | 150 | Deformable, quilted fabric |
| | plant | Pick & Place | 300 | Deformable, organic |
| | stuffed toy | Pick & Place | 300 | Deformable, soft fabric |
| | pouch | Pick & Place | 300 | Deformable, fabric, zipper |
| | scrubber | Pick & Place | 300 | Deformable, knitted yarn |

## A.6. Generating Textual Descriptions

We require short and long textual descriptions of each personal object to construct hard and short prompt baselines. To obtain these descriptions, we use GPT-5.1 with the following prompts (for long and short each). For each object, we use all reference views and query the model once.

---

**Prompt Used to Generate Short Textual Description for Hard Prompt Methods**

Describe only the object in the image. Focus on its most distinctive features such as shape, color, and material. Write the result in English as a short noun phrase, not a full sentence. Do not mention orientation, position, or surroundings.

---

**Prompt Used to Generate Long Textual Description for Hard Prompt Methods**

Describe only the object in the image. Write the result in English as a single, highly detailed noun phrase, not a full sentence. Include as many intrinsic and identifying features as possible, such as overall shape, dimensions, proportions, material, surface finish, textures, patterns, colors, edges, rims, openings, and any decorative or structural details. Do not mention orientation, position, or surroundings.

---

We use the short descriptions as hard prompts and the long descriptions as rich prompts for our personalization baselines. Table 11 in the appendix lists the final short and long descriptions for objects in Personalized-SIMPLER, Personalized-VLABench, and the real-world scenarios.

*Table 11.* Short and long textual descriptions for personal objects used in our benchmarks. Short descriptions and long descriptions are used as rich prompts for personalization baselines (hard prompt).

| Object | Short description | Long description |
|---|---|---|
| | *Personalized-SIMPLER* | |

| Object | Short description | Long description |
|---|---|---|
| **pen holder** | black cylindrical mesh pen holder | a cylindrical, black metal mesh pen holder with a smooth, solid circular base, featuring a fine grid pattern on the sides, a slightly flared top rim, and a matte finish |
| **bottle** | black cylindrical bottle with a red cap | a small, cylindrical black bottle with a smooth matte finish, featuring a colorful label with text and graphics, a slightly rounded cap, and a compact, uniform structure |
| **shaver** | a sleek black electric shaver with a curved handle and three rotary blades | a sleek, compact electric shaver with a matte black finish, featuring a slightly curved ergonomic handle for a comfortable grip, a glossy black control panel with subtle button details, and a rounded shaving head with three circular, silver metallic rotary blades encased in a dark gray protective mesh |
| **camera** | a small, black, cylindrical camera with a glossy finish | a small, black, rectangular camera with a cylindrical lens protruding from the front, featuring a glossy finish, smooth surfaces, subtle horizontal grooves on the body, a silver ring around the lens, and a small, circular button on the top |
| **owl figurine** | small, dark, textured owl ornament | a small, intricately carved owl figurine with a rounded body and prominent, detailed facial features, crafted from a dark, glossy material with subtle metallic sheen, featuring textured feather patterns across its surface, large circular eyes with a reflective finish, and a smooth, flat base for stability |
| **straw cup** | pink and white striped cup with a gray straw | a cylindrical cup with a tapered base, featuring a smooth surface adorned with alternating pink and white diagonal stripes, topped with a flat, circular lid with a central opening for a slender, straight, gray straw |
| *Personalized-VLABench* | | |
| **leather bag** | dark brown leather briefcase with a buckle clasp | a compact, rectangular, dark brown leather briefcase with a smooth surface finish, featuring a single flap closure secured by a metal clasp, a sturdy handle attached to the top, and subtle stitching along the edges for reinforcement |
| **shoe** | blue and gray athletic shoe with black laces and a chunky sole | a sleek athletic shoe with a rounded toe, featuring a combination of blue and gray synthetic materials, a mesh upper for breathability, a cushioned collar and tongue, a lace-up closure with black laces, a textured rubber outsole for traction, and subtle decorative stitching along the sides |

| Object | Short description | Long description |
|---|---|---|
| **cat figurine** | small black plush cat | a small, sleek black cat with a glossy coat, compact body, rounded ears, and a curled tail |
| **miniature house** | red and green toy house with a flat roof | a small, rectangular, toy-like house with a red roof, red walls, and blue windows, featuring a green base and a simple, blocky design |
| **cup** | a dark brown, cylindrical paper cup | a cylindrical, dark brown paper cup with a smooth matte finish, medium height, slightly tapered sides, a thin rolled rim, and an open top |

|  | *Real-world scenarios* |  |
|---|---|---|
| **vase** | clear glass vase with a narrow neck and bulbous base | a clear glass vase with a tall, narrow neck and a wide, bulbous base, featuring a smooth, glossy surface with subtle vertical ridges, a slightly flared rim, and a transparent finish that reveals the intricate reflections and refractions within |
| **plushie** | brown and pink fuzzy hedgehog plushie with a round body and pointed snout | a round, fuzzy plushie resembling a hedgehog with a soft, felt-like texture, featuring a light brown body and a darker brown, fluffy mane encircling its back, a protruding snout with a small, rounded brown nose, tiny black eyes, and subtle, rounded ears |
| **cup** | yellow ceramic cup with a cute animal face design | a round, ceramic cup with a glossy finish, featuring a wide, cylindrical body tapering slightly towards the top, adorned with a cute cartoon animal face design in shades of yellow, white, and pink, with large brown eyes, a small brown nose, a smiling mouth, and pink cheeks, complemented by a large, smoothly curved handle on the side |
| **slipper** | quilted beige slipper with brown trim | a beige quilted slipper with a closed-toe design, featuring a soft, padded upper with a subtle diamond pattern, a smooth tan fabric trim around the opening, a cushioned insole with a matching diamond pattern, and a stitched edge along the sole |
| **plant** | green succulent with thick, fleshy leaves | a compact, rosette-shaped succulent plant with thick, fleshy, elongated leaves exhibiting a smooth, matte surface and a muted green color, featuring slightly pointed tips and subtle, natural variegation along the edges |

| Object | Short description | Long description |
|---|---|---|
| **stuffed toy** | round plush toy with a beige face, red nose, and brown hat | a round, plush stuffed toy with a soft, tan-colored surface, featuring a circular face with embroidered black eyes, a small red nose, and a smiling black mouth, accented by light pink embroidered cheeks, and adorned with a light brown, textured hat-like rim encircling the top |
| **pouch** | small round beige fabric pouch with zipper and matching strap | a small soft-sided light gray rounded zippered pouch with a slightly flattened oval profile, finely ribbed synthetic fabric front panel, thick padded binding around the perimeter, continuous zipper track along the rim with braided-edge seam detail, gold-toned metal zipper hardware, and an attached matching woven webbing wrist strap loop with reinforced stitching and metal ring connector |
| **scrubber** | red and green crocheted strawberry-shaped scrubber | a handmade, strawberry-shaped scrubber featuring a vibrant red body with a chunky, knitted texture, accented by a contrasting bright green top resembling leaves, complete with a small loop for hanging, crafted from thick, soft yarn with a slightly fuzzy surface finish |

## B. Token Learning Strategy

In addition to VAP, we consider a token-based personalization strategy as a natural competitor, inspired by recent work on personalizing vision–language models and image generation models. In this line of work, a new token embedding is learned to represent a user-specific concept, while keeping the backbone model frozen. Methods such as MyVLM (Alaluf et al., 2024) and Yo'LLaVA (Nguyen et al., 2024) follow this paradigm and have shown strong personalization performance in question answering by learning concept tokens with carefully constructed hard and easy negatives.

Directly applying this idea to VLAs, however, is non-trivial. Yo'LLaVA optimizes its concept tokens using a text-based loss on downstream QA, whereas a VLA outputs continuous actions rather than text. To train concept tokens end-to-end for a VLA, one would need to collect personalized manipulation episodes for each user object, which introduces substantial additional data collection overhead. Instead, we adopt a proxy strategy: we first learn concept tokens in a frozen VLM following the Yo'LLaVA recipe, and then transfer the learned token embeddings into the VLA. As we show in the following subsections, the token embedding space changes little between the VLM and the corresponding VLA, making this transfer feasible in practice. We therefore treat token learning as a strong and principled baseline rather than our primary solution, and use the rest of this section to detail how we train the VLMs, analyze the token embedding space, and study the resulting attention patterns.

### B.1. Learning Token Embeddings on the Backbone VLM

We closely follow the training pipeline of Yo'LLaVA (Nguyen et al., 2024) when learning personalized tokens in the VLM. Concretely, we start from a frozen PaliGemma 3B model (Beyer et al., 2024), which combines a SigLIP image encoder with a Gemma-based language decoder and is designed for transfer to downstream vision–language tasks. For each personal object, we introduce a new subject identifier token (e.g., `<my_cup>`) into the vocabulary together with a small number of learnable soft tokens. The rest of the VLM parameters, including the vision encoder and language decoder, remain fixed during training.

The personalized token is optimized on a recognition-style VQA dataset rather than on long conversational QA. For each

personal object $o$, we construct yes/no questions of the form *"Can you recognize <my_cup> in this image?"* and train the VLM to output a short answer (*"yes"* or *"no"*). This design follows the observation that PaliGemma tends to perform best on short VQA-style prompts and succinct answers; we found that longer conversational templates often led to unstable behavior and mode collapse.

Positive examples use the user-provided reference images of $o$. For hard negatives, instead of mining from a large external dataset as in Yo'LLaVA (Nguyen et al., 2024), we leverage the actual distractors that appear in our personalized benchmarks. For each distractor instance, we capture more than ten views and label them as hard negatives for the corresponding object token. Easy negatives are sampled from objects that never appear as distractors for $o$ (e.g., props from other tasks), providing a diverse background of unrelated images. This choice is favorable to the token-learning baseline and can be viewed as a form of evaluation-set leakage, since the hard negatives are drawn directly from the deployment-time distractor instances. We adopt this setting intentionally to strengthen the baseline; thus, the improvements of VAP over Soft Prompt should be interpreted as conservative.

Unlike Yo'LLaVA, which represents each subject as a sequence of several learnable soft tokens (e.g., <my_cup> is <token_0><token_1>...<token_k>), we observed that long learned prefixes do not work well with PaliGemma: as $k$ grows, the prompts deviate from the model's pretraining distribution and the answers become unstable. We therefore learn a single special token embedding for each personal object and place it directly in the question, as in *"Can you recognize <my_cup> in this image?"*. During training we freeze all VLM parameters and optimize only this token embedding using a standard cross-entropy loss on the yes/no answers. The resulting embedding lies in the PaliGemma token space and is later copied into the corresponding token embedding matrix of the VLA.

Following Yo'LLaVA, we train each personalized token for 20 epochs using the same optimization hyperparameters as the original method. We monitor performance on a held-out recognition QA set constructed in the same way as the training questions and answers. After training, the VLM attains over 95% accuracy on this validation split for all personal objects, indicating that the learned token reliably captures the visual identity of each object in the VLM's embedding space.

### B.2. Token Embedding Space Analysis

Our proxy strategy relies on the assumption that the token embedding space of the VLM remains largely unchanged when the model is fine-tuned into a VLA. To verify this, we compare the token embeddings of PaliGemma 3B (used as our VLM) with those of the corresponding VLA ($\pi_0$ and $\pi_{0.5}$), which is obtained by training on large-scale robotic action data (Intelligence et al., 2025). Both models share the same tokenizer and vocabulary, so their embeddings are aligned by construction.

We extract the token embedding matrices from the text decoders of PaliGemma 3B and each VLA variant, and evaluate their alignment using three metrics: token-wise cosine similarity, linear centered kernel alignment (CKA), and a k-nearest-neighbor top-1 (kNN@1) token matching accuracy. CKA measures how similar the pairwise inner-product structure of two representation spaces is, and equals 1 when one space can be obtained from the other by an orthogonal transformation. The kNN@1 score measures, for each VLM token, whether its nearest neighbor in the VLA embedding space is the corresponding token, thus directly quantifying how often token identities are preserved.

As summarized in Table 12, both $\pi_0$ variants are essentially identical to the VLM embeddings: they achieve a mean cosine similarity of $0.999999 \pm 0.000000$, a CKA of $1.000000$, and a kNN@1 of $0.9998$, indicating that almost every token in the VLM embedding matrix is mapped to its exact counterpart in the VLA. The $\pi_{0.5}$ variants still exhibit a high mean cosine similarity of $0.945626 \pm 0.066187$, a CKA of $0.985684$, and kNN@1 scores between $0.9899$ and $0.9913$, suggesting that the global geometry of the token embedding space is largely preserved and that token-wise correspondences remain highly stable despite the additional action fine-tuning. Together, these metrics support our approximation that concept tokens learned on the PaliGemma VLM can be transferred to the corresponding VLA without relearning the embedding space from scratch, as they continue to live in essentially the same token embedding space.

These observations are consistent with prior findings on language model fine-tuning, which report that lower layers and embedding matrices change little compared to upper layers during task-specific adaptation (Merchant et al., 2020; Hao et al., 2020). In our context, they suggest that learning personalized tokens on the PaliGemma VLM and then transferring them to the VLA is a reasonable approximation: the token embeddings live in essentially the same space before and after action fine-tuning. This motivates our use of VLM-trained concept tokens as a strong token-learning baseline for personalized manipulation.

*Table 12.* Token embedding space comparison between PaliGemma-3B (VLM) and various VLA models. All metrics show that the token embedding space remains largely unchanged after fine-tuning the VLM into a VLA, with near-perfect cosine similarity and minimal L2 distance.

| Model | Vocab Size | Cosine Similarity | CKA | kNN@1 |
|---|---|---|---|---|
| $\pi_0$ (Personalized-SIMPLER Google Robot) | 257152 | $0.999999 \pm 0.000000$ | 1.000000 | 0.9998 |
| $\pi_0$ (Personalized-SIMPLER WidowX) | 257152 | $0.999999 \pm 0.000000$ | 1.000000 | 0.9998 |
| $\pi_{0.5}$ (Personalized-VLABench) | 257152 | $0.945626 \pm 0.066187$ | 0.985684 | 0.9913 |
| $\pi_{0.5}$ (Real-world Scenario) | 257152 | $0.945626 \pm 0.066187$ | 0.985684 | 0.9899 |

### B.3. Attention Analysis

In the previous subsections we showed that token learning can successfully fit concept tokens in the VLM and that these tokens transfer almost unchanged into the VLA's embedding space. A natural question is whether the learned token actually attends to the correct personal object inside the VLA. To probe this, we visualize the spatial attention induced by the learned concept token on the VLA's visual features.

Concretely, we extract the VLA's cross-attention weights for the query position corresponding to the learned object token, and take the attention mass over image-token keys as a per-patch score from the last layer (layer 17). We then reshape and resize these scores to the image resolution and normalize them across spatial locations to obtain a heatmap. Intuitively, this heatmap measures how strongly the concept token "recognizes" each region of the image. Figures 9–12 show representative examples from Personalized-SIMPLER, overlaid on the original RGB observations.

Across these examples, the learned token *often* assigns higher responses to the personal object than to background regions, indicating that it can bias the VLA's visual representations toward the user-specified instance. However, we also observe that this localization is frequently *fragile* over time: as the rollout progresses and the robot's viewpoint and scene configuration change, the peak activation can shift across frames, sometimes drifting to nearby objects that share similar shape, texture, or category-level cues. Even when the heatmap remains relatively consistent on the target for several steps, the model still exhibits a tendency to select or interact with an incorrect object in cluttered scenes, suggesting that token-induced localization alone does not provide stable, instance-level tracking under continuous, closed-loop execution.

Consistent with our main experiments (Section 5), this instability helps explain why the token-learning baseline does not reliably improve end-to-end manipulation. In particular, the baseline may (i) briefly emphasize the correct object but then drift to a distractor later in the episode, or (ii) maintain reasonable emphasis on the target yet still execute an action on a distractor or fail due to manipulation difficulty. Overall, learned tokens can partially influence where the model "looks," but the signal is often time-varying and insufficiently discriminative to robustly resolve visually similar instances throughout a multi-step manipulation trajectory.

One plausible explanation for the remaining failures, including episodes where the heatmap appears relatively consistent on the target yet the behavior is still incorrect, is the VLA architecture. The vision–language encoder that produces the text/image embeddings (and contains the learned concept token) is shared with the underlying VLM, whereas the action head is a separate expert trained via flow matching. During action prediction, latent action variables are sampled from noise and attend over the entire sequence of text and image embeddings through cross-attention, mixing information across modalities and tokens. As a result, even when the learned token induces a stable preference in the perceptual representations, this fine-grained, instance-specific cue can be diluted or overridden before it consistently affects the final low-level action distribution. This suggests that, beyond improving token-induced localization, bridging token-level conditioning into reliable instance-aware control in a separate action expert remains challenging. While our analysis focuses on the $\pi$-family backbone used in this paper, similar modular designs with a language/vision trunk and a separate action head are common in contemporary VLAs, so this limitation may extend beyond this specific backbone.

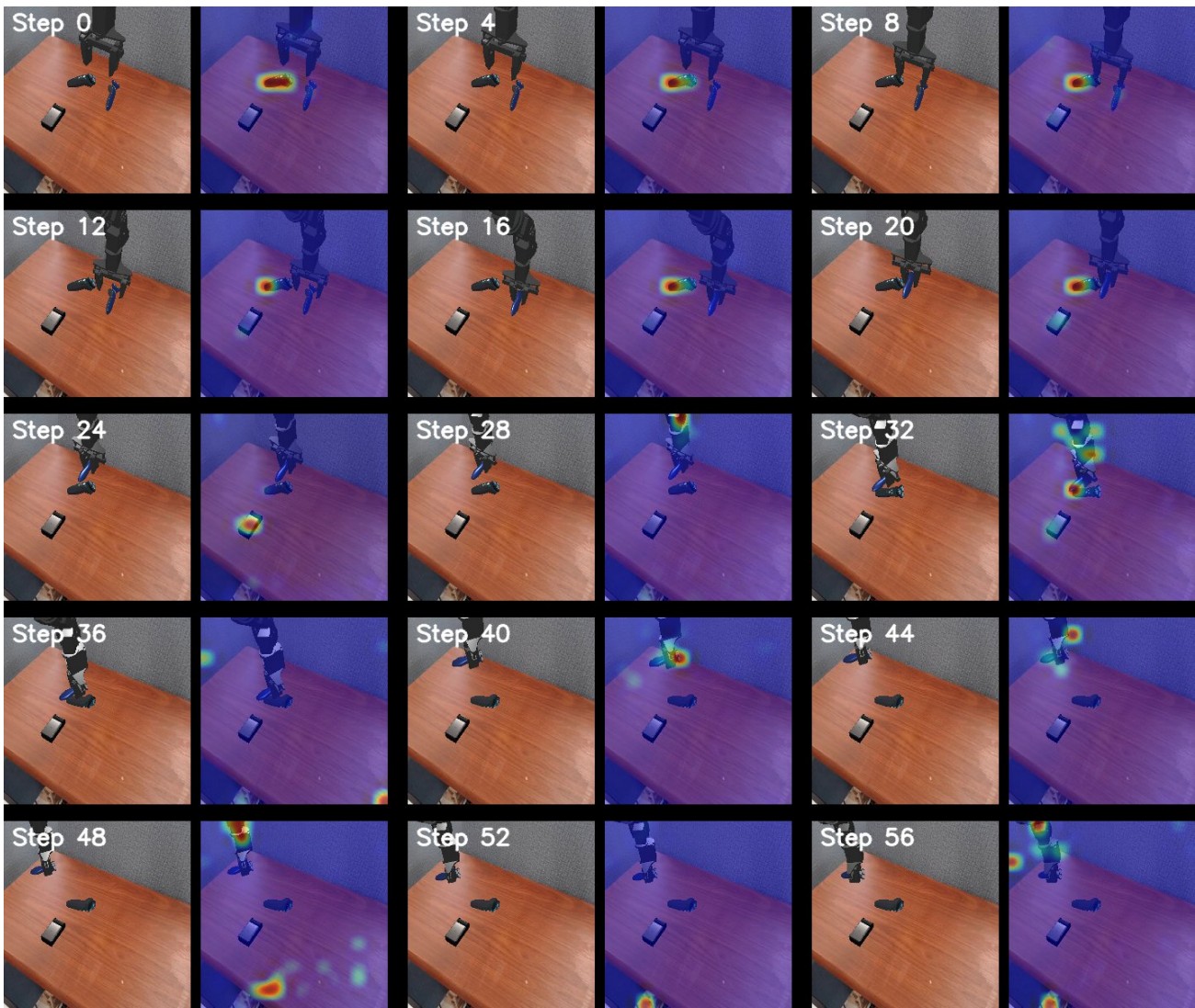

*Figure 9.* **Soft Prompt: relatively consistent localization yet failed execution.** Across the rollout, the token–patch similarity heatmaps remain largely concentrated near the intended personal object, but the episode still fails (e.g., incorrect interaction or failure to complete the manipulation), illustrating that correct or stable localization does not necessarily translate into successful control.

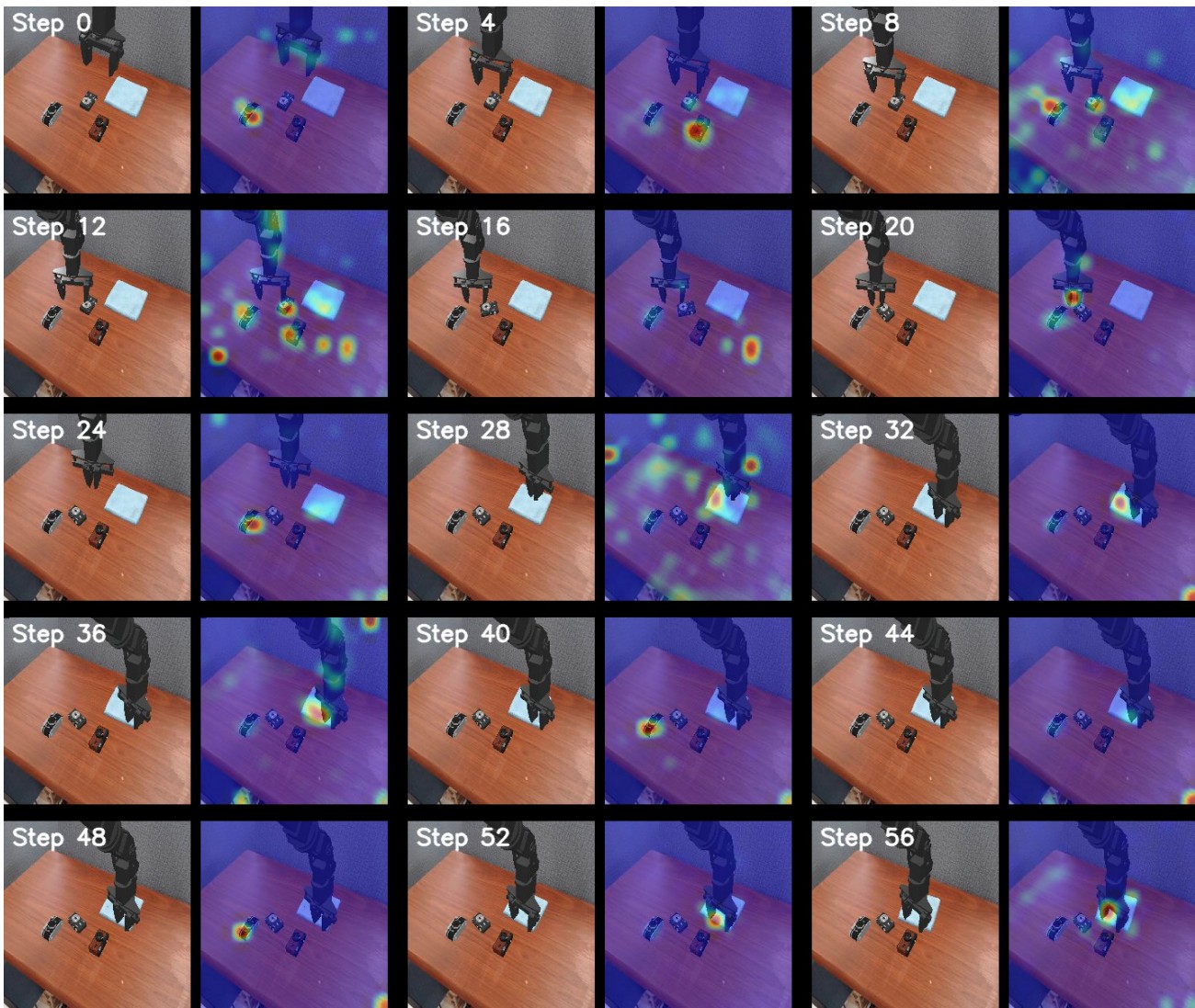

*Figure 10.* **Soft Prompt: temporally unstable localization.** As the robot moves and the viewpoint changes, the heatmap peaks shift across frames and intermittently activate on different objects/background regions, indicating that the token-induced preference can be fragile over time during closed-loop execution.

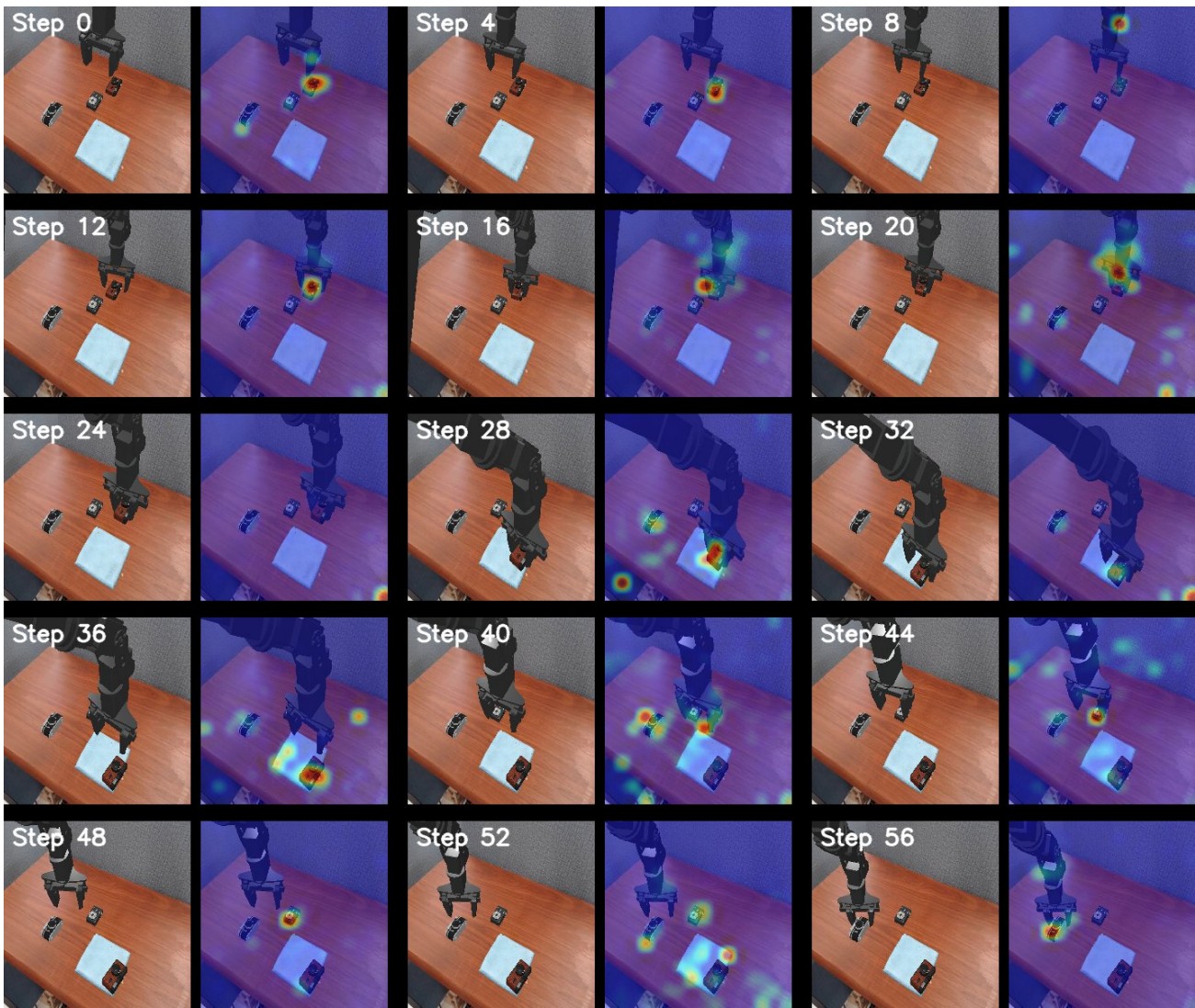

*Figure 11.* **Soft Prompt: confusion under visually similar distractors.** In the presence of near-identical instances, the heatmaps show substantial activation on distractors in addition to (or instead of) the target, suggesting that a learned token often provides only a coarse instance bias and may not reliably disambiguate visually similar objects.

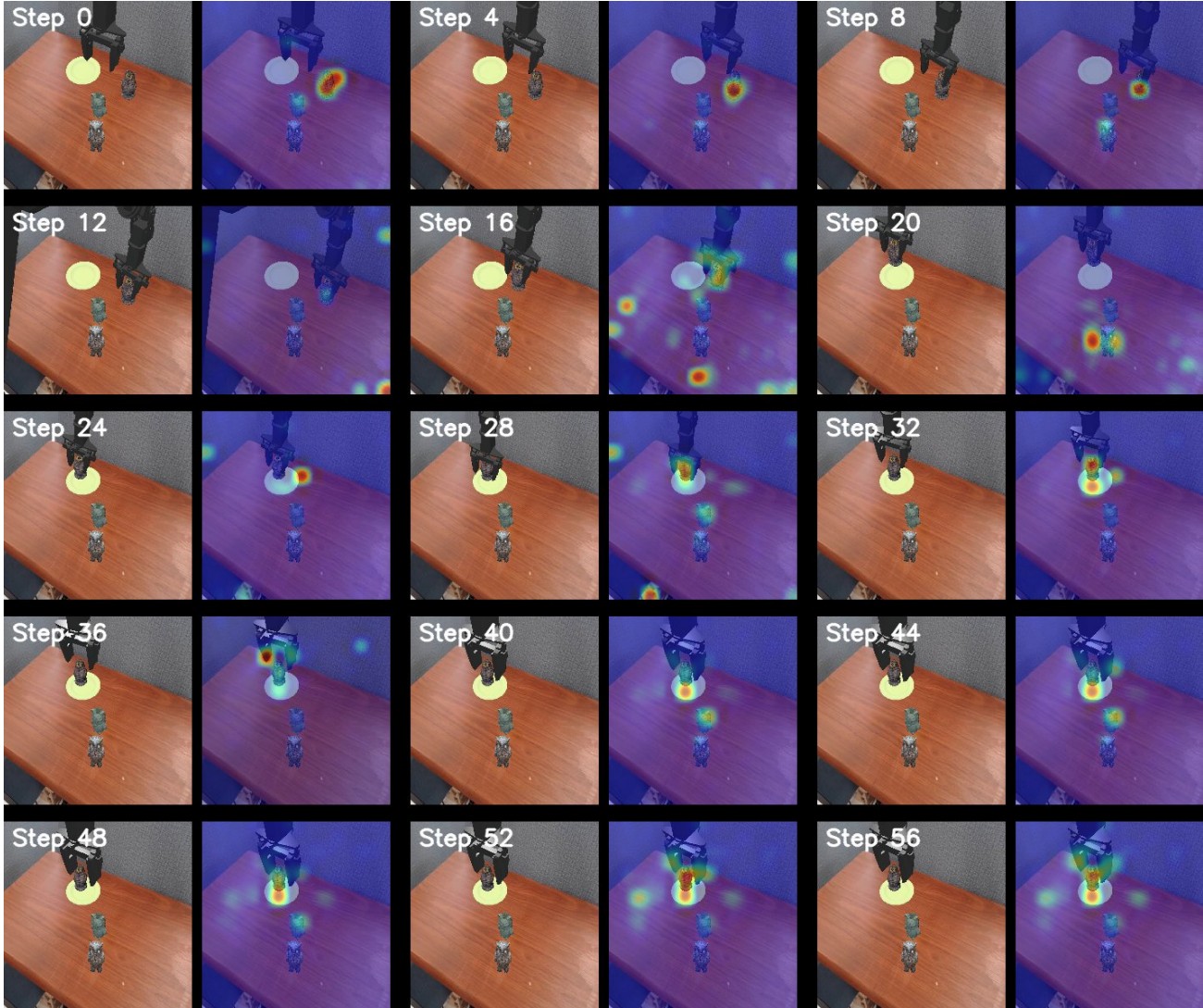

*Figure 12.* **Soft Prompt: a successful episode.** This example shows a case where the heatmaps place comparatively higher responses on the intended personal object across the rollout and the task succeeds, highlighting that token learning can sometimes provide useful instance bias, albeit not consistently across scenes and time.

## C. Qualitative Examples

In this section, we present additional qualitative examples that illustrate how VAP behaves in practice. For each task in each benchmark, we show representative rollouts where the agent successfully identifies and manipulates the user's personal object among visually similar distractors. For each rollout, we visualize all frames of the episode, including the generated visual prompts overlaid on the input images.

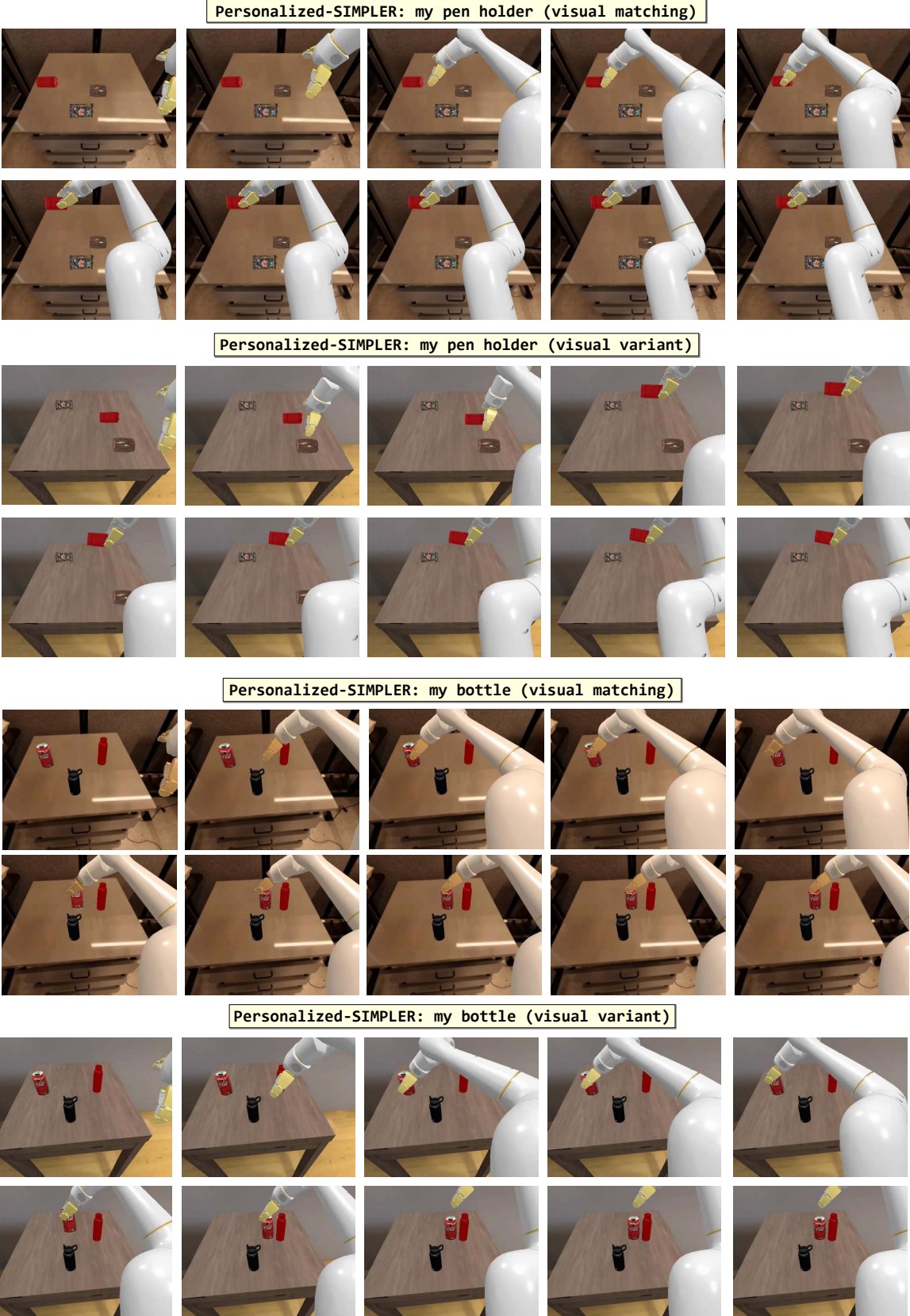

*Figure 13.* Qualitative Examples on Personalized-SIMPLER (Google Robot)

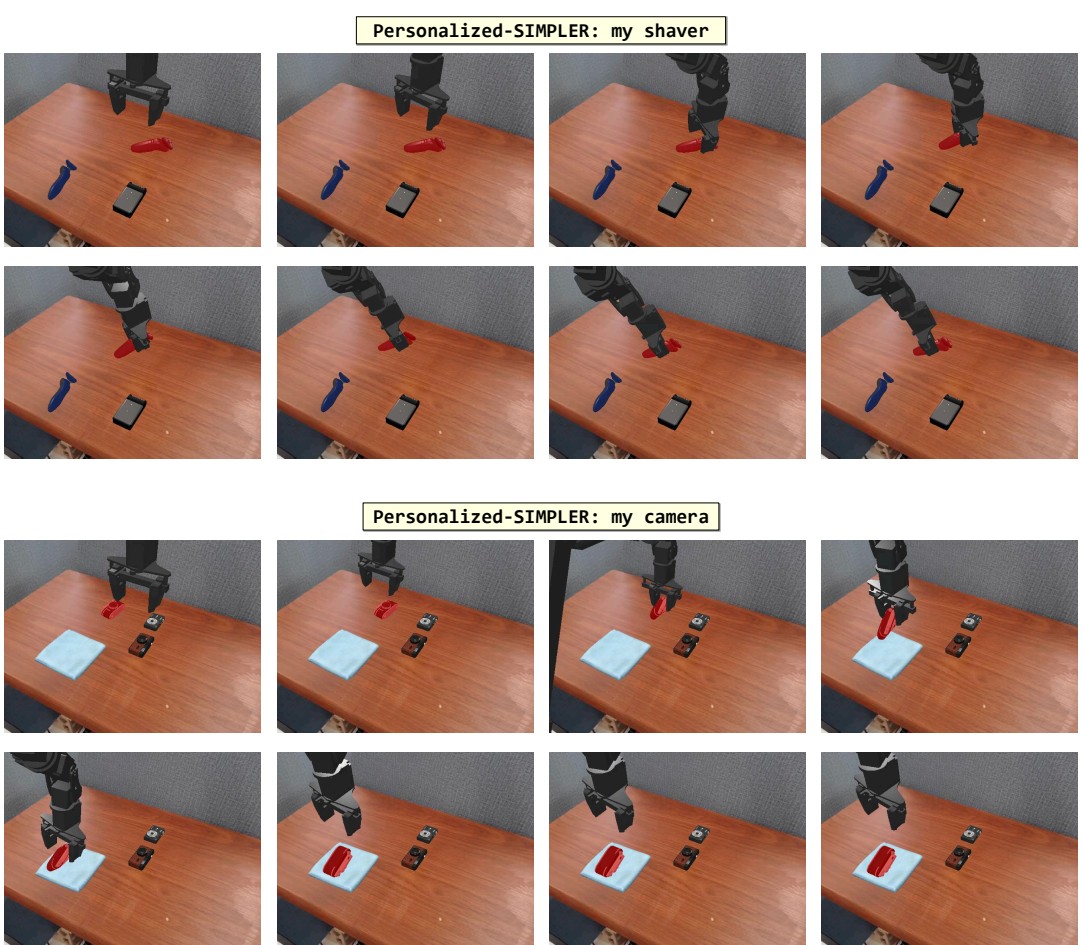

*Figure 14.* Qualitative Examples on Personalized-SIMPLER (WidowX) - 1

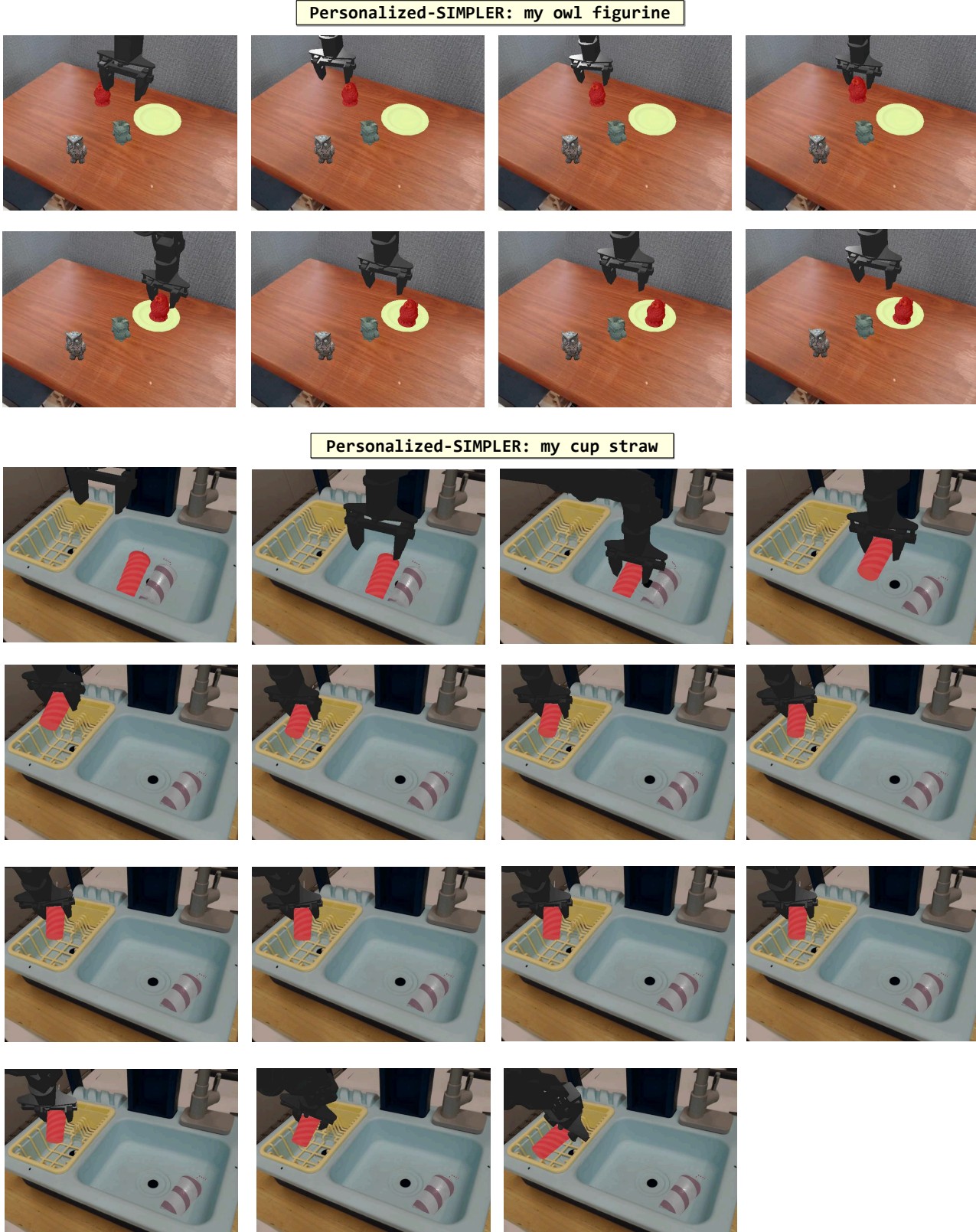

Figure 15. Qualitative Examples on Personalized-SIMPLER (WidowX) - 2

Personalized-VLABench: my leather bag

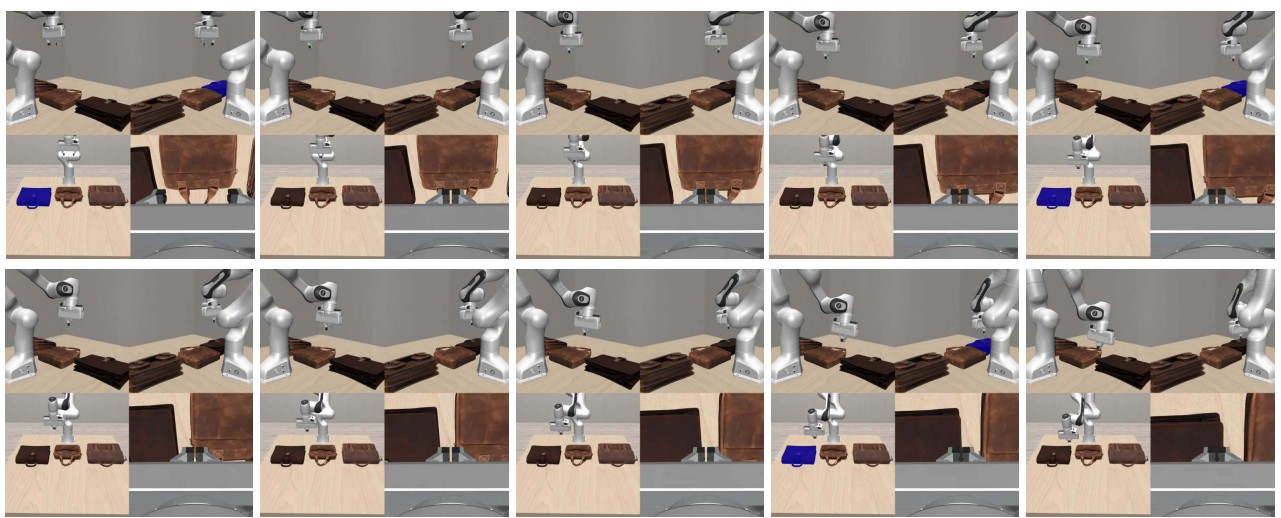

Personalized-VLABench: my shoe

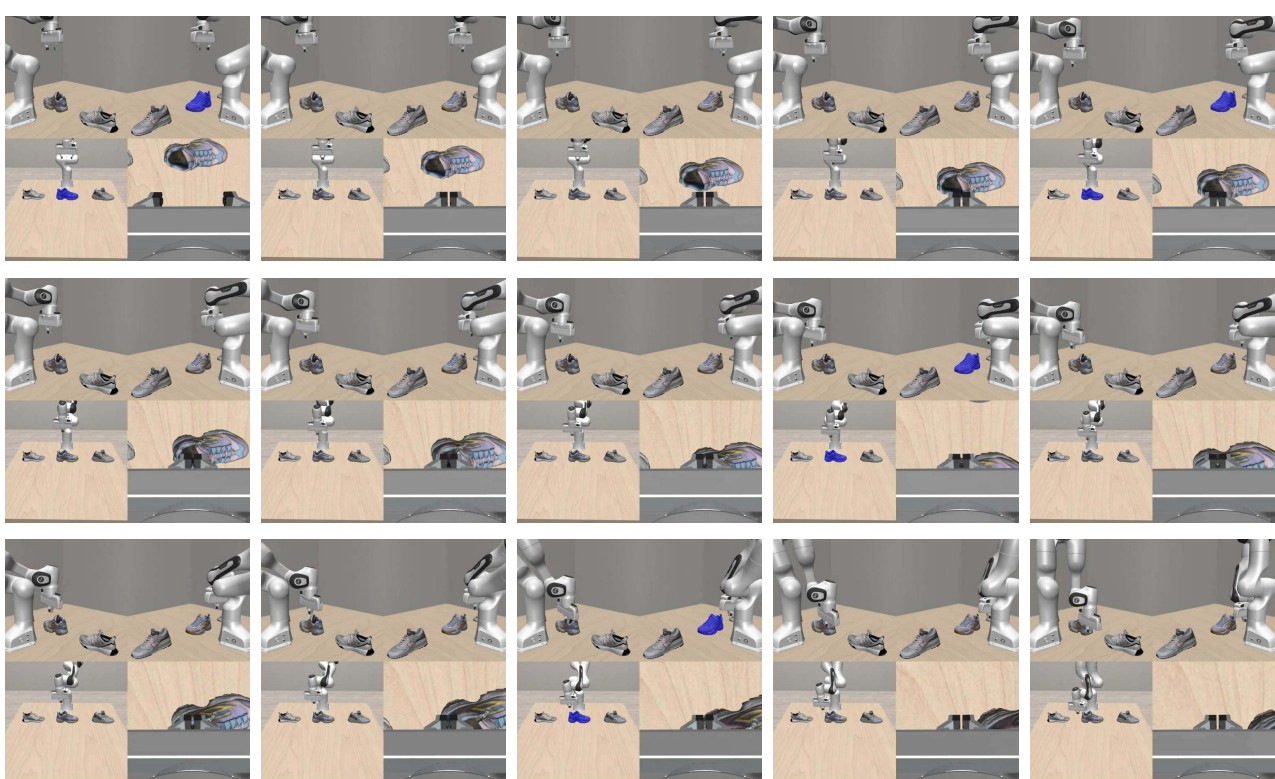

*Figure 16.* Qualitative Examples on Personalized-VLABench - 1

Personalized-VLABench: my cat figurine

Personalized-VLABench: my miniature house

Personalized-VLABench: my cup

*Figure 17.* Qualitative Examples on Personalized-VLABench - 2

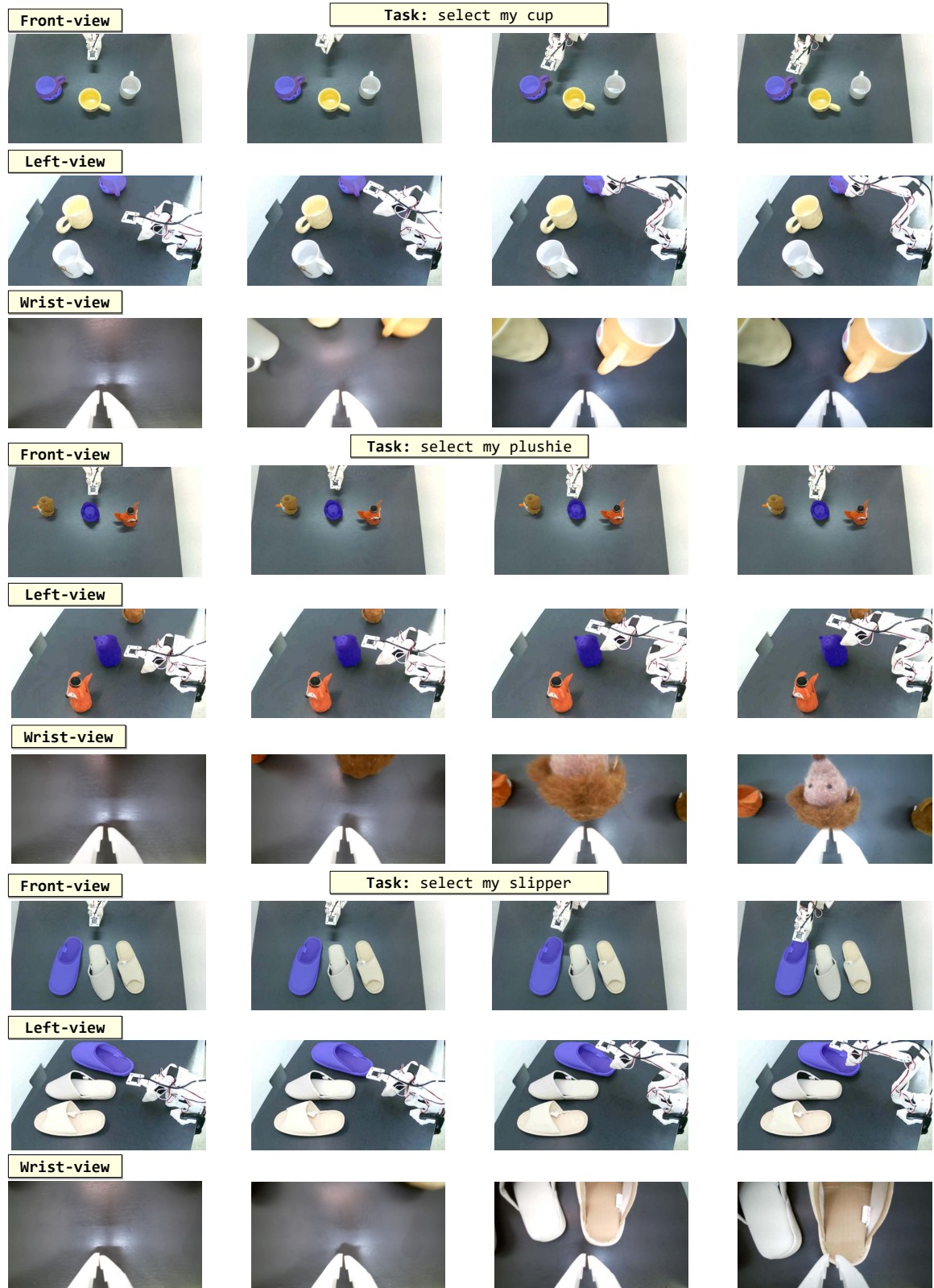

*Figure 18.* Qualitative Examples on Real-world Scenario - 1

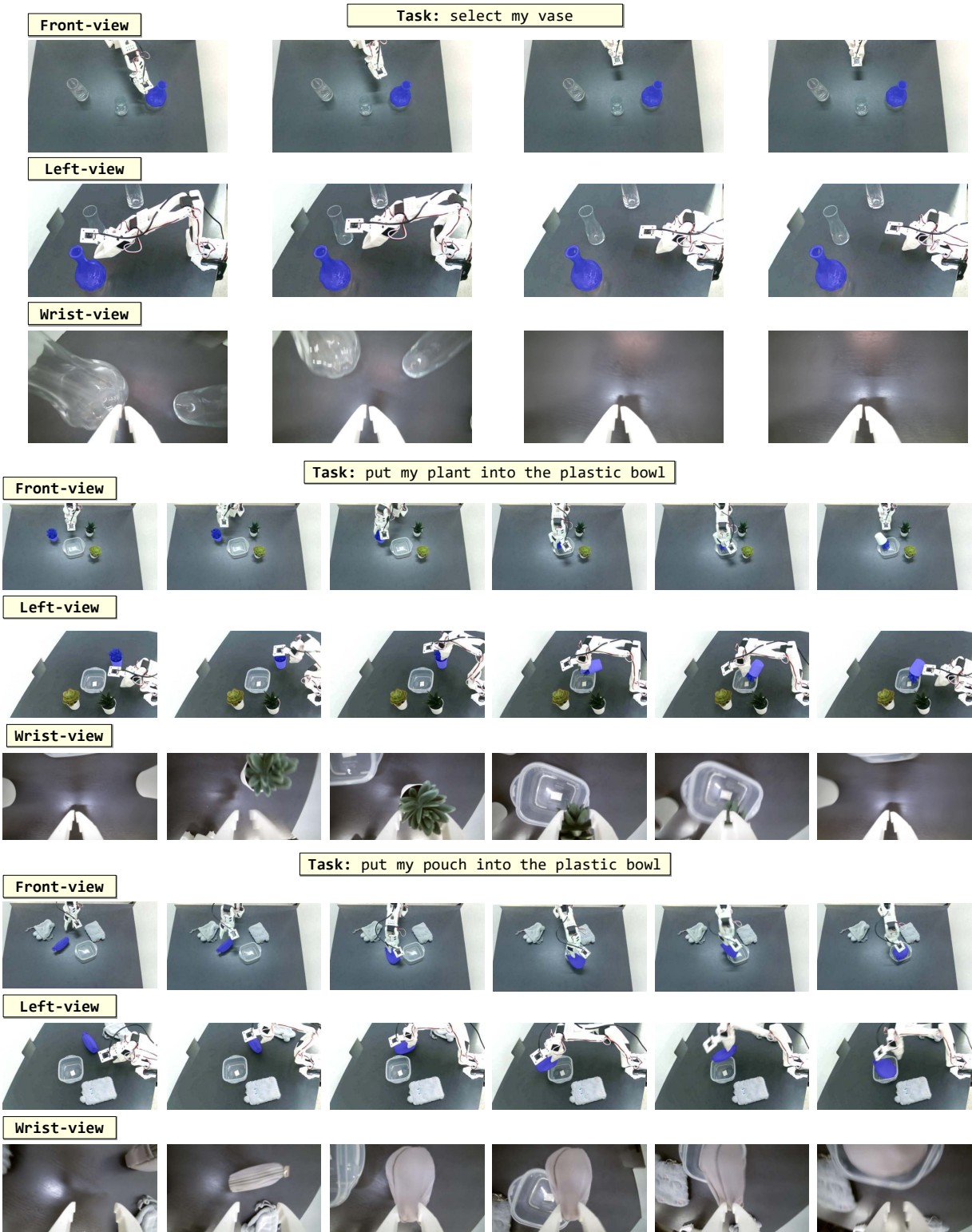

*Figure 19.* Qualitative Examples on Real-world Scenario - 2

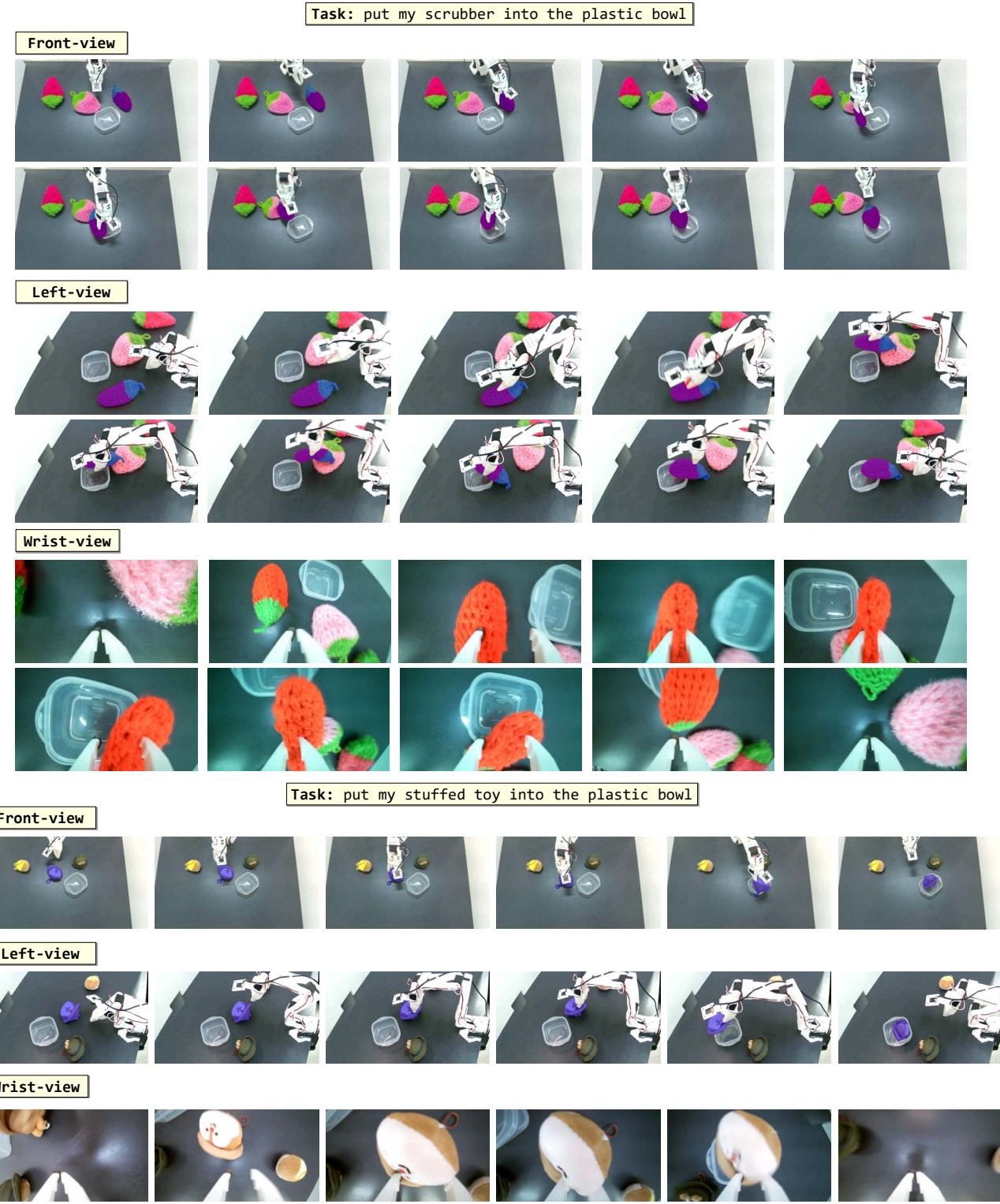

*Figure 20.* Qualitative Examples on Real-world Scenario - 3

## D. Error Case Analysis

While VAP substantially improves personalized manipulation success, it still fails in certain characteristic ways. In this section, we categorize typical failure modes and provide qualitative examples for each. Beyond illustrating representative rollouts, these cases help disentangle whether failures primarily stem from the upstream perception/prompting pipeline or from the downstream VLA behavior under challenging manipulation dynamics. As summarized in Table 13, failures in single-view settings are less often due to wrong prompts (Case 1: 13.1%/21.4% of failures in Fractal/Bridge), whereas multi-view settings are dominated by cross-view inconsistencies (Case 2: 59.2%/50.9% of failures in Personalized-VLABench/Real-world). We group errors into three types: (1) incorrect visual prompts arising from single-view ambiguity, (2) inconsistent prompts across multiple views, and (3) correct prompts that nonetheless do not lead to successful actions from the VLA. Figures 21–23 show representative examples for each category; they are intended to illustrate the taxonomy rather than provide an exhaustive survey of failures.

### D.1. Case 1: Wrong Visual Prompt (Single View)

In the first category, the perception pipeline produces an incorrect mask for the personal object even though only a single camera view is involved (13.1%/21.4% of failures in Fractal/Bridge; Table 13). We observe two common sub-cases: (i) the open-vocabulary detector incorrectly localizes the *generic* class in a cluttered scene (e.g., producing an imprecise or shifted box/mask that already excludes the true instance), and (ii) the open-vocabulary detection is reasonable but the subsequent image-encoder matching assigns a higher similarity score to a distractor than to the ground-truth personal instance (i.e., the top-1 retrieval is wrong). The latter can occur even when the target is visible, particularly when multiple near-identical instances exist and the reference set does not capture the most discriminative view. In such cases, the selected visual prompt can remain stable yet consistently highlight the wrong object throughout the episode; the failure is therefore rooted in the grounding function $g$, and the VLA simply and consistently acts on the object it is shown. This also suggests a concrete direction for improvement: adapting or training the image encoder to better separate *personalized* instances (e.g., instance-discriminative matching) could reduce retrieval-induced prompt errors without changing the downstream VLA (see Figure 21).

### D.2. Case 2: Wrong Visual Prompt (Multi-View Inconsistency)

The second category arises from inconsistencies across camera views and accounts for a large fraction of failures in multi-view settings (59.2% in Personalized-VLABench and 50.9% in Real-world; Table 13). Our current system factorizes grounding and tracking per camera view for real-time inference: retrieval/matching and mask propagation are performed independently in each view, and the resulting per-view masks are directly used as prompts. Under severe occlusion, partial visibility (especially in the wrist camera), or extreme viewpoint changes, this factorized design can cause per-view matching/tracking to lock onto different same-category instances across cameras. Consequently, one camera may correctly highlight the personal object while another highlights a distractor, yielding contradictory prompts that can destabilize behavior or bias actions toward the wrong view/object.

Note that the voting mechanism (Sec. 3.3) aggregates evidence across *reference images* within each view to select the target proposal. However, it does not couple camera views or enforce a shared cross-view identity during grounding, and thus cannot prevent per-view drift under occlusion or eliminate contradictory prompts across cameras. These cases expose a limitation of our current independent per-view design and motivate future work on explicitly enforcing cross-view consistency.

**Future direction: cross-view association for joint evidence fusion (MAP-style).** Case 2 fundamentally stems from the lack of an explicit *shared* target identity across views: our current pipeline performs retrieval/matching and tracking independently per camera, so there is no latent instance variable that couples observations across cameras. A natural extension is therefore to introduce an intermediate cross-view association stage that enforces a single physical instance identity across views *before* generating per-view prompts, and then aggregates multi-view evidence while keeping the downstream VLA policy frozen.

Formally, let each view $v \in \{1, \dots, V\}$ produce candidate instance proposals $\mathcal{B}^{(v)} = \{b_i^{(v)}\}_{i=1}^{M_v}$ (e.g., detector masks/boxes with short-horizon tracks) with proposal embeddings $e_i^{(v)}$ (already computed for reference matching), and let the reference set $R_o$ provide embeddings $\{z_r\}_{r \in R_o}$. We represent the shared identity by a per-view assignment vector $m = \{m_v\}_{v=1}^V$ where $m_v \in \{1, \dots, M_v\} \cup \{\varnothing\}$ selects which proposal corresponds to the target in view $v$ (or $\varnothing$ when the target is

unobserved due to occlusion). A MAP-style discrete approximation is to solve

$$m^* = \arg \max_m \underbrace{\sum_{v=1}^{V} \phi_{\text{ref}}(v, m_v)}_{\text{reference evidence}} + \lambda \underbrace{\sum_{v<u} \phi_{\text{cv}}(v, m_v;\, u, m_u)}_{\text{cross-view consistency}} + \beta \underbrace{\sum_{v=1}^{V} \phi_{\text{obs}}(v, m_v)}_{\text{visibility/track prior}},$$

where $\phi_{\text{ref}}$ measures reference agreement (e.g., $\phi_{\text{ref}}(v, i) = \sum_{r \in R_o} \cos(e_i^{(v)}, z_r)$ and $\phi_{\text{ref}}(v, \varnothing) = 0$), $\phi_{\text{obs}}$ encodes optional unary priors (e.g., detector confidence or track stability), and $\phi_{\text{cv}}$ encourages that selected proposals across views correspond to the same physical instance. In the following, we keep the unary terms $\phi_{\text{ref}}$ and $\phi_{\text{obs}}$ fixed and discuss alternative instantiations/approximations for $\phi_{\text{cv}}$ and the resulting joint inference. Below we outline a few plausible design directions for cross-view association, rather than an exhaustive treatment.

**Embedding-based clustering.** A simple instantiation sets $\phi_{\text{cv}}(v, i;\, u, j) = \cos(e_i^{(v)}, e_j^{(u)})$ (and 0 if either side is $\varnothing$), reusing already-computed proposal embeddings. A common approximation is to build a cross-view graph over proposals with edge weights given by $\phi_{\text{cv}}$, cluster proposals into cross-view instance hypotheses $\mathcal{K}$, and then select the cluster maximizing aggregated unary evidence, $\text{score}(k) = \sum_{(v,i) \in k} \big( \phi_{\text{ref}}(v, i) + \beta \, \phi_{\text{obs}}(v, i) \big)$. Per-view prompts are then produced from members of the chosen cluster, enforcing agreement by construction.

**Bipartite matching and track consistency.** Another standard approximation solves pairwise assignments between view pairs. For a pair $(v, u)$, one can define a cost that combines appearance and optional geometry (when available), e.g.,

$$\text{cost}(i, j) = -\cos(e_i^{(v)}, e_j^{(u)}) + \gamma \, d_{\text{geom}}\Big( b_i^{(v)}, b_j^{(u)} \Big),$$

and apply Hungarian matching to obtain consistent correspondences, which can then be merged into multi-view tracklets/clusters followed by the same multi-view evidence aggregation.

**Geometry-aware coupling when calibration is available.** When camera calibration and robot kinematics are available, geometric consistency can further instantiate $\phi_{\text{cv}}$ via reprojection constraints. For example, letting $c_i^{(v)}$ denote a proposal centroid/keypoints in view $v$ and $\Pi_u(\cdot)$ the projection into view $u$, one can use a coarse 3D proxy $\hat{X}_i$ (e.g., triangulation from multiple views, a depth prior, or kinematic constraints) and define

$$\phi_{\text{cv}}(v, i;\, u, j) = \cos(e_i^{(v)}, e_j^{(u)}) - \eta \left\| \Pi_u\Big( \hat{X}_i \Big) - c_j^{(u)} \right\|_2 \quad \text{(or IoU between a projected region and } b_j^{(u)} \text{)}.$$

This formalizes the intuition that calibration/kinematics can couple views through reprojection, but it is sensitive to calibration/synchronization and the quality of the 3D proxy and may introduce new failure modes if applied naively.

**Learned cross-view correspondence as a building block.** Alternatively, learned cross-view correspondence/segmentation modules (e.g., ObjectRelator) may provide building blocks to associate per-view proposals across ego-/exo-centric perspectives and enforce agreement (Fu et al., 2025). In the above formulations, such modules can instantiate $\phi_{\text{cv}}$ via learned association scores (or a soft correspondence field), which can be particularly helpful when geometric cues are weak or unavailable.

**Why this is non-trivial (expected challenges).** While the objective is conceptually clean, naively adding cross-view association can introduce a new bottleneck. Cross-view matching is itself error-prone under heavy occlusion and extreme viewpoint changes, and erroneous associations can be worse than leaving views independent by forcing all prompts to agree on an incorrect identity. Moreover, proposal embeddings may not be sufficiently view-invariant for near-identical instances, clustering/assignment can fragment a true instance or merge multiple instances, and geometry-based coupling depends on calibration accuracy and reliable 3D proxies. Finally, joint inference increases computation and debugging complexity, requiring careful approximations and uncertainty handling to maintain real-time performance. We therefore leave explicit cross-view association and joint evidence fusion as important directions for future work.

### D.3. Case 3: Correct Visual Prompt but Failed Manipulation

In the third category, the visual prompt correctly highlights the personal object in all relevant views, yet the agent still fails the manipulation task (Case 3 accounts for 86.9%/78.6% of failures in Fractal/Bridge and 40.8%/49.1% in Personalized-VLABench/Real-world; Table 13). Qualitatively, we often observe that the robot approaches the visually prompted object

(indicating that the prompt successfully resolves instance identity), but the episode still fails due to manipulation difficulty(e.g., tight grasps, clutter-induced collisions, or precise placement requirements where small execution errors accumulate). In these examples, improving grounding alone is unlikely to close the remaining gap: success appears limited by the VLA's underlying manipulation capability and robustness under contact-rich, cluttered dynamics. This points to a complementary direction, namely strengthening the VLA itself (e.g., better base policy competence or robustness), to convert correct instance conditioning into reliably successful end-to-end manipulation (see Figure 23).

*Table 13.* Error-mode breakdown for VAP across benchmarks with more details. "Fail (%)" is computed over all evaluation episodes. Case 1–3 are computed conditional on failure and should sum to 100% within each row. Case 2 (multi-view inconsistency) is not applicable to single-view settings and Case 1 (single-view ambiguity) is not applicable to multi-view settings.

| Benchmark | Setting | #Eps | Fail (%) | Case 1 (%) | Case 2 (%) | Case 3 (%) |
|---|---|---|---|---|---|---|
| Personalized-SIMPLER (Fractal) | Google Robot (1 view) | $1,685$ | 37.6 | 13.1 | — | 86.9 |
| Personalized-SIMPLER (Bridge) | WidowX (1 view) | $96 \times 10$ | 16.6 | 21.4 | — | 78.6 |
| Personalized-VLABench | Franka (3 views) | $125 \times 10$ | 38.4 | — | 59.2 | 40.8 |
| Real-world | SO-101 (3 views) | $20 \times 8$ | 31.9 | — | 50.9 | 49.1 |

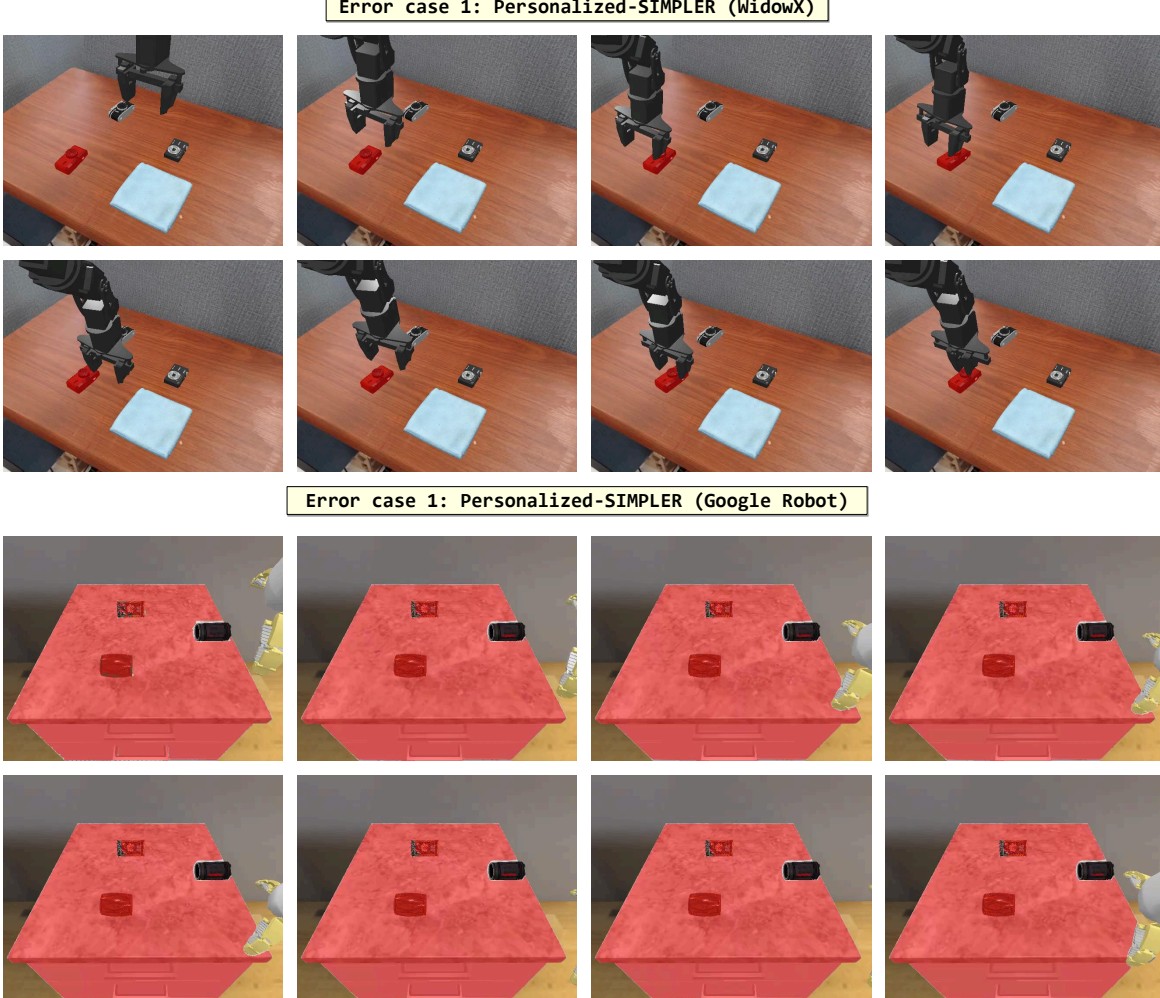

*Figure 21.* **Case 1 (single view): wrong visual prompt due to grounding/matching error.** The open-vocabulary detector proposes a plausible region for the generic class, but the instance-level matching selects a distractor whose embedding similarity exceeds that of the true personal object; the resulting mask consistently highlights the wrong instance, and the robot correspondingly interacts with the distractor.

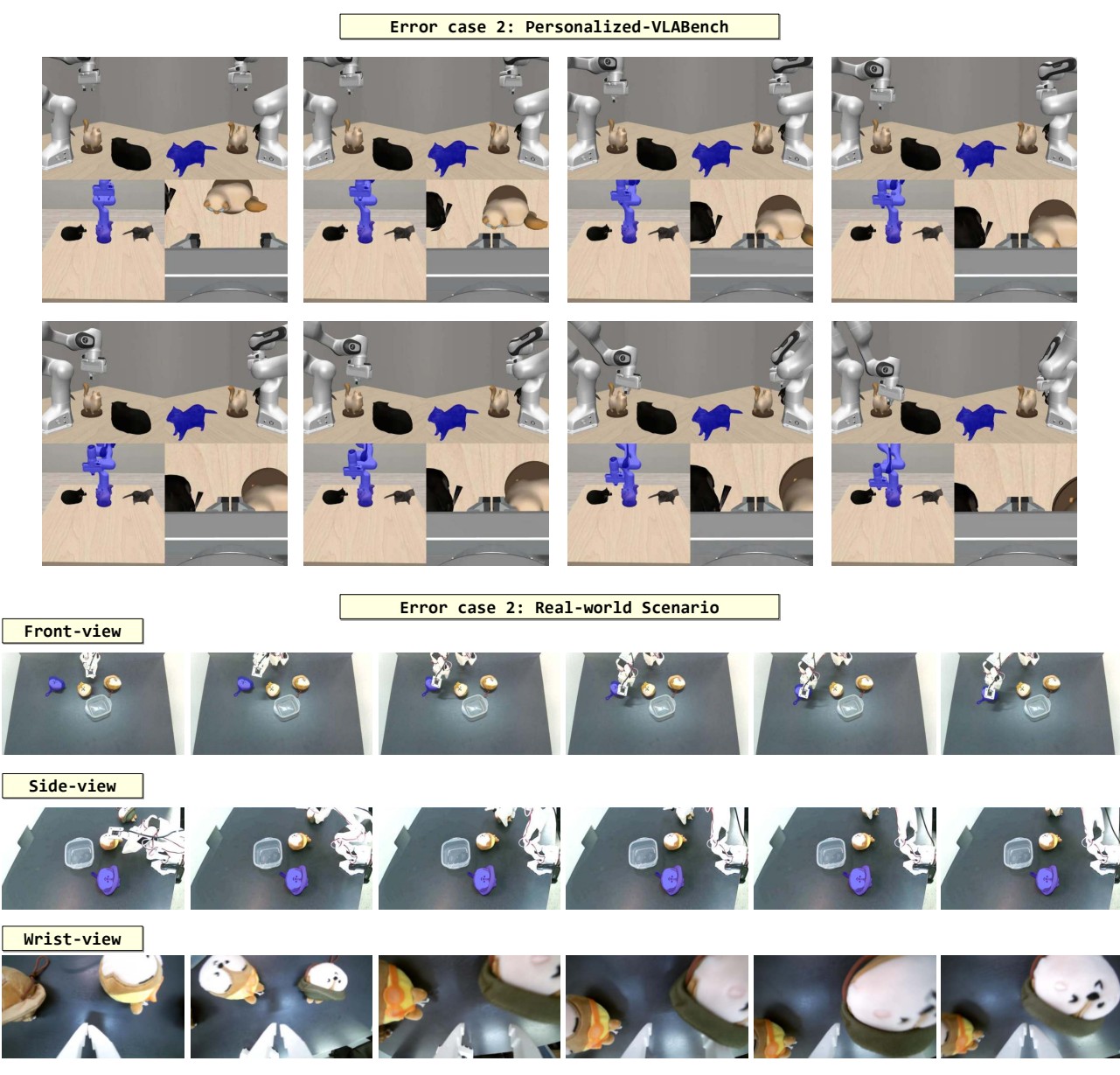

*Figure 22.* **Case 2 (multi-view): inconsistent visual prompts across cameras.** Per-view grounding/tracking locks onto different instances in different views (e.g., the target in one camera but a distractor in another), yielding contradictory mask prompts; this cross-view inconsistency leads to unstable or biased actions toward the wrong view/object.

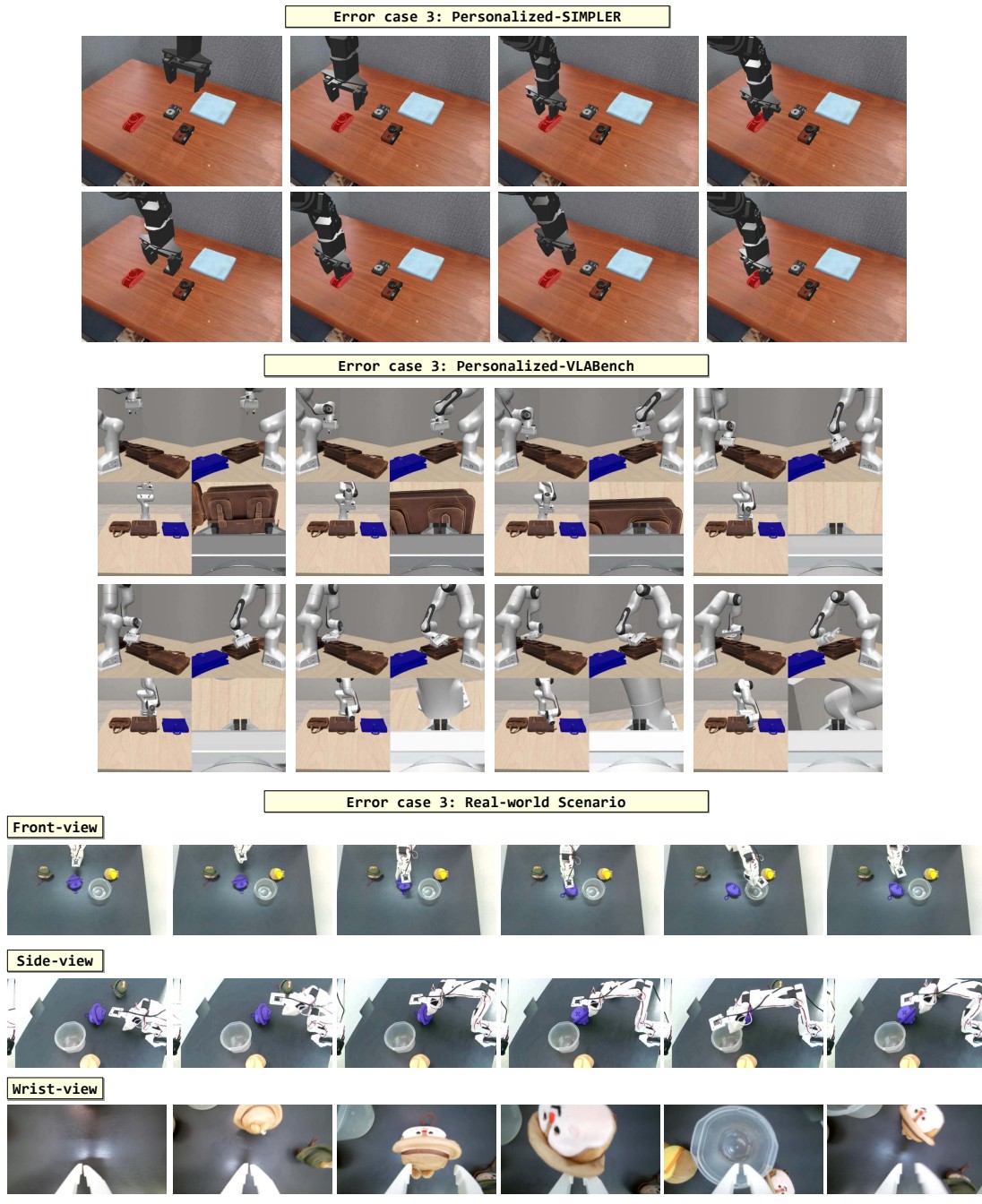

*Figure 23.* **Case 3 (correct prompt): correct instance highlighted but manipulation fails.** The mask prompt consistently highlights the intended personal object in all relevant views, and the robot approaches the prompted object, yet the episode fails due to challenging contact-rich manipulation (e.g., tight grasps, clutter-induced collisions, or precise placement tolerances), suggesting a limitation of the underlying VLA policy rather than grounding.

*Table 14.* Ablation of instruction rewriting on single-view Personalized-SIMPLER. "Mask-only" removes rewriting, while "Rewrite-only" removes the visual highlight but keeps the same tint-color rewrite template, serving as a negative control for language-only cueing.

| Method | Personalized-SIMPLER (Google Robot) | | Personalized-SIMPLER (WidowX) | |
|---|---|---|---|---|
| | CMR (%) ↑ | SR (%) ↑ | CMR (%) ↑ | SR (%) ↑ |
| VAP (mask + rewrite) | **83.7** | **62.7** | **93.6** | **83.5** |
| Mask-only | 20.4 | 7.3 | 23.9 | 5.2 |
| Rewrite-only | 35.2 | 15.8 | 41.7 | 18.8 |

# E. Ablation Studies

This section ablates key design choices in Visual Attentive Prompting (VAP) to clarify which components are responsible for the gains reported in Section 5. We conduct all ablations on Personalized-SIMPLER to isolate the effects of grounding and prompt design under a controlled visual setting. Multi-view benchmarks introduce an additional confound, namely cross-view association and consistency, that can dominate failure cases (Section 5.5). We therefore analyze multi-view inconsistency separately and focus here on component-level causality.

Unless otherwise stated, all variants use the same frozen VLA policy and perception backbones as VAP, and follow the same evaluation protocol and metrics (Section 5). Tables 14–17 summarize the results.

### E.1. Rewriting the Instruction: Is It Necessary?

VAP combines a visual highlight with a lightweight instruction rewrite (Section 3.3) to align the language referent with the prompted region. To test whether rewriting is a critical ingredient, we compare three variants: *(i)* VAP (mask + rewrite), *(ii)* mask-only (visual highlight with the original instruction unchanged), and *(iii)* rewrite-only (rewritten instruction without any visual highlight). Table 14 reports performance on Google Robot and WidowX in Personalized-SIMPLER.

**Results and Analysis.** Table 14 shows that combining the visual highlight with instruction rewriting is essential for strong performance. VAP achieves 62.7% SR / 83.7% CMR on Google Robot and 83.5% SR / 93.6% CMR on WidowX, whereas removing rewriting (Mask-only) collapses performance to 7.3%/20.4% (SR/CMR) and 5.2%/23.9%, respectively. This indicates that a visual marker alone is not reliably exploited by the frozen VLA when the instruction remains in a personalized form (e.g., "my X"). Instead, language–vision alignment via rewriting is a critical ingredient. Conversely, rewriting without any visual prompt (Rewrite-only) provides only limited gains. Note that our rewrite template explicitly introduces the tint-color word (e.g., "the red $c$") even when the highlight is absent; thus, Rewrite-only serves as a *negative control* that tests whether a color-based linguistic cue alone can drive personalization in a cluttered scene. The limited performance indicates that the gains of VAP do not come from language rewriting in isolation, but from the coupled language–vision alignment created by *both* the mask highlight and the rewrite.

### E.2. Aggregating Evidence Across References: Voting vs. Averaging

In the grounding function $g$, VAP selects the target proposal by aggregating reference-view similarity via a voting rule (Section 3.3). A natural alternative is to average similarity scores across references and pick the maximum. We compare these two aggregation schemes while keeping all other components fixed. Table 15 summarizes the results.

**Results and Analysis.** Table 15 compares two aggregation rules for selecting the target proposal across reference images. Voting yields consistently stronger results than averaging, with modest gains on Google Robot (83.7 vs. 82.0 CMR; 62.7 vs. 62.3 SR) and more pronounced gains on WidowX (93.6 vs. 87.4 CMR; 83.5 vs. 80.6 SR). While both rules perform competitively, the consistent improvement motivates our default choice. Beyond accuracy, voting provides an *interpretable consensus* across reference views: each reference contributes a discrete top-1 preference, and the final selection is the proposal supported by the majority, which aligns naturally with our non-parametric reference-memory framing and simplifies diagnosis of grounding decisions.

*Table 15.* Ablation of reference aggregation on single-view Personalized-SIMPLER. "Voting" is the default rule in VAP (Section 3.3), while "Averaging" selects proposals by mean cosine similarity across references.

| Aggregation | Personalized-SIMPLER (Google Robot) | | Personalized-SIMPLER (WidowX) | |
|---|---|---|---|---|
| | CMR (%) ↑ | SR (%) ↑ | CMR (%) ↑ | SR (%) ↑ |
| Voting (VAP) | **83.7** | **62.7** | **93.6** | **83.5** |
| Averaging | 82.0 | 62.3 | 87.4 | 80.6 |

*Table 16.* Sensitivity to the number of reference images $K$ on single-view Personalized-SIMPLER. We subsample the reference set and report SR/CMR for Google Robot and WidowX.

| $K$ | Personalized-SIMPLER (Google Robot) | | Personalized-SIMPLER (WidowX) | |
|---|---|---|---|---|
| | CMR (%) ↑ | SR (%) ↑ | CMR (%) ↑ | SR (%) ↑ |
| $K = 1$ | 60.2 | 45.6 | 73.7 | 61.9 |
| $K = 3$ | 81.2 | 59.4 | 90.2 | 78.4 |
| $K = 5$ | 83.7 | 62.7 | **93.6** | 83.5 |
| $K = 7$ | **84.1** | **62.8** | 93.1 | **85.2** |

### E.3. Sensitivity to the Number of Reference Images

VAP uses a small set of reference images as a non-parametric memory (Section 3.1). To quantify sensitivity to reference availability, we subsample the reference pool and evaluate VAP with $K \in \{1, 3, 5, 7\}$ reference images.

**Results and Analysis.** Table 16 shows a clear benefit from increasing the number of reference images. With a single reference ($K=1$), VAP already performs reasonably on both robots, indicating a practical low-reference regime. Increasing to $K=3$ yields the largest improvement, consistent with the value of additional viewpoint coverage for distinguishing near-identical instances. Moving from $K=3$ to $K=5$ provides further gains, while increasing to $K=7$ yields only marginal additional improvement, suggesting that performance begins to saturate with a small handful of references. Based on this trade-off, we use $K=5$ as the default setting throughout the paper, capturing most of the benefit while keeping the reference collection burden and initialization cost modest.

### E.4. Visual Prompt Design: Geometry, Highlight Strength, and Comparison to Prompting Baselines

Finally, we ablate the visual prompt design used to inject grounding into the frozen VLA (Section 3.3). We evaluate two aspects: *(i)* the prompt geometry, comparing a segmentation-mask tint (default in VAP) against a bounding-box-based prompt, and *(ii)* the highlight strength, varying the overlay opacity $\alpha$. We additionally compare VAP against a broader set of visual prompting strategies adopted in prior robotics work, including opaque masks (RoboGround), points, circles, trajectory arrows (TraceVLA), and numbered markers (PIVOT, MOKA). Table 17 summarizes all three factors.

**Results and Analysis.** Table 17 highlights that both the *geometry* and *strength* of the visual prompt matter. First, segmentation-mask highlighting consistently outperforms bounding-box prompting at the same opacity, suggesting that instance-aligned prompts provide a cleaner and less ambiguous attention signal for manipulation. In contrast, bounding boxes often include background and nearby distractors, which can dilute the conditioning and interfere with fine-grained action execution. Second, the opacity sweep reveals a saliency–occlusion trade-off: very weak highlights are insufficient to reliably guide the policy, while overly strong overlays can obscure visual cues relevant for precise grasping and placement. Overall, intermediate opacity values perform best, motivating our default choice ($\alpha=0.5$) used in VAP.

Among the alternative prompting strategies, mask-aligned identity tinting yields the strongest results, outperforming RoboGround-style opaque masks ($\alpha=1.0$) by 14–18 SR points, bounding-box prompting by 28–36 points, and point, circle, trajectory-arrow (TraceVLA), and numbered-marker (PIVOT, MOKA) prompts by 35–40 SR points on the harder WidowX setting. Spatial- and affordance-style prompts were designed for per-step grounding rather than for maintaining persistent instance identity; when multiple same-category lookalikes are present, per-step re-grounding triggers identity switches and cascading failures, while identity-based tinting preserves instance binding across frames.

*Table 17.* Ablation of visual prompt design on single-view Personalized-SIMPLER. The top block compares VAP's mask-aligned tint against alternative visual prompting strategies, including approaches adopted in prior robotics work (RoboGround, TraceVLA, PIVOT, MOKA). The bottom block sweeps the overlay opacity $\alpha$.

| Prompt Variant | Personalized-SIMPLER (Google Robot) | | Personalized-SIMPLER (WidowX) | |
|---|---|---|---|---|
| | CMR (%) ↑ | SR (%) ↑ | CMR (%) ↑ | SR (%) ↑ |
| Mask (VAP, $\alpha = 0.5$) | **83.7** | **62.7** | **93.6** | **83.5** |
| Mask ($\alpha = 1.0$, RoboGround) | 76.5 | 48.2 | 86.0 | 65.8 |
| Box ($\alpha = 0.5$) | 54.0 | 33.9 | 65.6 | 47.2 |
| Point | 42.0 | 20.5 | 52.8 | 32.0 |
| Circle | 48.5 | 26.8 | 60.2 | 40.5 |
| Trajectory arrow (TraceVLA) | 38.5 | 18.2 | 48.0 | 28.5 |
| Numbered marker (PIVOT, MOKA) | 45.0 | 24.0 | 56.5 | 36.8 |
| Mask ($\alpha = 0.1$) | 43.2 | 22.5 | 47.9 | 26.0 |
| Mask ($\alpha = 0.3$) | 75.2 | 52.4 | 80.5 | 73.2 |
| Mask (VAP, $\alpha = 0.5$) | **83.7** | **62.7** | 93.6 | **83.5** |
| Mask ($\alpha = 0.7$) | 80.1 | 61.5 | **94.2** | 83.2 |

*Table 18.* Controlled occlusion sweep on Personalized-SIMPLER. We vary the number of consecutive frames during which the target is fully occluded and report tracking accuracy and task success rate. The spatio-temporal tracker maintains identity through several seconds of complete invisibility.

| Frames | Personalized-SIMPLER (Fractal, 0.75 Hz) | | | Personalized-SIMPLER (Bridge, 1.25 Hz) | | |
|---|---|---|---|---|---|---|
| | Duration | Track (%) ↑ | SR (%) ↑ | Duration | Track (%) ↑ | SR (%) ↑ |
| 0 | 0.00s | 99.8 | 62.7 | 0.00s | 98.0 | 83.5 |
| 1 | 1.33s | 99.5 | 62.2 | 0.80s | 97.5 | 83.0 |
| 2 | 2.67s | 98.2 | 61.0 | 1.60s | 96.5 | 82.0 |
| 3 | 4.00s | 95.5 | 58.5 | 2.40s | 94.0 | 79.5 |
| 4 | 5.33s | 89.8 | 53.2 | 3.20s | 88.5 | 74.2 |
| 5 | 6.67s | 80.2 | 45.0 | 4.00s | 79.0 | 65.0 |
| 6 | 8.00s | 65.5 | 35.8 | 4.80s | 64.0 | 52.5 |

**Robustness of the Tint Cue.** Three aspects of the tint design contribute to robust identity binding. First, the tint provides a consistent identity cue across frames, which the policy reliably attends to among visually similar instances (Tables 2–4). Second, performance is stable across $\alpha$=0.3–0.7 and degrades only at the fully opaque regime ($\alpha$=1.0) that VAP explicitly avoids, so underlying texture and semantic cues used for manipulation remain visible beneath the overlay. Third, tint colors are selected to maximize contrast against the scene palette, and we did not observe systematic failures attributable to color collision in any of our benchmarks.

### E.5. Tracking Robustness Under Occlusion

A common concern with mask-based identity binding is whether the system remains robust when the target temporarily disappears from view, a frequent occurrence during manipulation as the gripper or arm occludes the object. To quantify this, we vary the duration of complete target invisibility and report both the fraction of episodes in which the tracker correctly re-associates the target upon reappearance (Track) and the resulting success rate (SR). Occlusion durations are reported in seconds, derived from the observation rate of each platform (0.75 Hz Fractal / 1.25 Hz Bridge due to action chunking).

**Results and Analysis.** Tracking accuracy remains above 95% when the target is occluded for up to 3–4 frames on both platforms (4.0 s Fractal / 2.4 s Bridge), with SR dropping less than 5 points relative to the no-occlusion baseline. Performance degrades gracefully thereafter, with the tracker still recovering the correct instance in roughly two-thirds of episodes after 8 s of continuous invisibility on Fractal. These results indicate that the spatio-temporal memory in our mask propagation step is the key mechanism that prevents single-view grounding errors from cascading into manipulation failures during typical manipulation-induced occlusion events.

### E.6. Sensitivity to Mask Quality

The accuracy of the SAM2 mask determines the spatial precision of the tint overlay. To stress-test sensitivity to mask quality, we apply morphological dilation (background bleed, ratio $> 1$) and erosion (partial coverage, ratio $< 1$) to the SAM2 output before tinting, and report the resulting SR on Personalized-SIMPLER (Table 19).

*Table 19.* Sensitivity to mask quality on Personalized-SIMPLER. Ratios above 1.0 dilate the mask, ratios below 1.0 erode it.

| Mask Ratio | Fractal SR (%) | Bridge SR (%) |
|---|---|---|
| 0.25 (extreme erosion) | 30.5 | 48.0 |
| 0.5 | 45.8 | 65.2 |
| 1.0 (baseline, SAM2 default) | **62.7** | **83.5** |
| 1.5 | 54.0 | 74.2 |
| 2.0 (extreme dilation) | 42.3 | 60.8 |

**Results and Analysis.** Performance is stable around the baseline mask quality: moderate dilation ($1.5\times$) and erosion ($0.5\times$) incur at most a 9–17 point SR drop, while standard SAM2-quality masks operate well within this stable regime. Extreme corruption ($0.25\times$ or $2.0\times$) causes larger degradation but is well outside the typical quality range of off-the-shelf segmentation models, indicating that VAP degrades gracefully under realistic perception noise.

## F. Practical Usage Scenarios

In practical deployments, multiple users often share a workspace and issue personalized requests that refer to different users' belongings (e.g., *"my brother's dog figurine"* vs. *"my ornament"*). While VAP is primarily an input-side personalization mechanism, it naturally extends to such shared-scene settings. Our key design choice is to execute multi-object requests *sequentially*, applying a *single* mask prompt at a time, which avoids overwhelming the frozen VLA with multiple concurrent visual cues. This section provides a small real-world demonstration of this capability; we do not claim that the VLA infers ownership from vision alone, but rather that VAP correctly binds ownership terms to visual prompts and executes the resulting sub-goals reliably.

**Setup.** We use the real-world SO-101 tabletop setup (Section 4.2). The scene contains two user-associated target objects(a `brother-dog` figurine and an `ornament`) along with 2 distractor objects from the same categories for each target. For each target, we collect $K=5$ reference images from the real cameras to form the corresponding reference set $R_o$.

**Unified Sequential Execution Strategy.** We consider a compound instruction involving both targets, e.g., *"put my brother's dog figurine into the plastic bowl, and then put my ornament into the plastic bowl."* We assume a lightweight high-level planner (e.g., rule-based decomposition or an off-the-shelf LLM) to convert the request into two ordered sub-goals. The system then executes these sub-goals sequentially to maximize reliability. For each sub-goal, it retrieves the corresponding reference set, applies the standard VAP pipeline to generate a *single* target mask (and corresponding instruction rewrite), and executes the action with the frozen VLA. After completing the first action, the mask is cleared and VAP is re-run on the updated scene to *re-ground* the second target, making the second action robust to state changes caused by the first manipulation.

**Metrics.** Each trial consists of two sub-tasks (brother's dog figurine, then ornament). We report: (i) **2-step Success Rate** (both sub-tasks succeed), (ii) **Wrong-Object Rate** (the robot manipulates a distractor object), and (iii) **Others** (all remaining failures, e.g., grasp/placement failures or collisions despite targeting the intended object).

**Results.** As shown in Table 20, the unified sequential strategy achieves a 65.0% 2-step success rate; among failures, 20.0% are due to manipulating distractors and 15.0% fall into *Others*. Overall, these results suggest that VAP can be readily used in shared-scene, multi-user settings by decomposing multi-object requests into sequential sub-goals with single-target prompting. In practice, this provides a simple and reliable path to extending VAP beyond single-object personalization without modifying the frozen VLA.

**Qualitative Examples.** Figure 24 visualizes representative rollouts, overlaying the VAP mask at each sub-task to make the grounding decision explicit. The figure highlights how the system focuses on one target at a time (brother's dog figurine

*Table 20.* Multi-user practical usage in a shared scene. We execute a two-object request sequentially with a single target mask at a time. Reported over 20 randomized trials with 2 distractor objects from the same categories for each target object.

| Method | 2-step SR (%) ↑ | Wrong-obj. (%) ↓ | Others (%) ↓ |
|---|---|---|---|
| Unified Sequential (Single-mask VAP) | 65.0 | 20.0 | 15.0 |

first, then ornament) and updates the mask after the first action changes the scene.

**Simultaneous Multi-Object Instructions.** Beyond sequential execution, VAP also supports *indecomposable* multi-object instructions in which two personal targets are referenced jointly within a single command (e.g., *"put my pouch near my vase"*). The grounding pipeline is run in parallel, assigning distinct color tints to each target and rewriting the instruction accordingly (e.g., *"put the red pouch near the blue vase"*), without any architectural changes or retraining. We evaluate five dual-object scenarios on the real-world tabletop setup (Section 4.2), each with 20 trials (100 trials total). As shown in Table 21, VAP reliably manipulates both personal objects within a single instruction.

*Table 21.* Dual-object instructions on the real-world tabletop setup. Each scenario assigns distinct color tints to both personal objects and rewrites the instruction accordingly. Reported over 20 trials per instruction (100 trials total).

| Instruction | Object 1 | Object 2 | SR (%) |
|---|---|---|---|
| Put my pouch near my vase | Pouch | Vase | 75.0 |
| Move my plushie next to my cup | Cup | Plushie | 65.0 |
| Put my scrubber near my slipper | Slipper | Scrubber | 70.0 |
| Move my stuffed toy next to my plant | Stuffed Toy | Plant | 60.0 |
| Put my pouch near my cup | Pouch | Cup | 70.0 |

# G. Discussion, Limitations, and Future Work

VAP is a training-free framework that personalizes pre-trained Vision–Language–Action models for user-specific robotic manipulation by augmenting them with a modular front-end for personal-object recognition and instruction rewriting. Across two simulation benchmarks and a real-world robotic arm, VAP substantially improves instance-level success over non-personalized and token-based baselines. We see this work as an input-side foundation that opens several natural extensions, with mobile manipulation being a particularly natural direction where active recovery from persistent occlusion and exploratory viewpoint changes become central to robust personalized assistance. Our current instantiation is scoped to tabletop scenes with a single personalized object and short-horizon tasks, and it relies on off-the-shelf open-vocabulary detection and segmentation to provide accurate masks. When grounding is incorrect or inconsistent across camera views, these errors propagate to the visual prompts and can lead to failures (Appendix D). We view these aspects not as fundamental obstacles, but as concrete axes along which VAP can be strengthened, for example, by incorporating stronger cross-view 3D grounding, lightly adapting the action head to better exploit visual prompts, and combining our module with planners that reason over sequences of personalized sub-tasks.

Even at the level of object manipulation, there is substantial room to expand the definition of personalization. It is not only about *which* object to handle, but also *how* to handle it: one user may prefer that their mug is always grasped by the handle, while another may require that a fragile plant is moved by supporting the pot from below. These preferences can be naturally modeled as user-specific affordance maps, conditioning not just the target selection but also contact locations and approach directions. Extending VAP to learn such fine-grained preferences from a small number of demonstrations is a promising direction for safer and more user-aligned manipulation.

Looking ahead, we envision personalization as a key enabler for long-term human-robot interaction. The principles of VAP could be adapted to personalize spatial concepts (e.g., learning what "my side of the desk" implies) or to execute context-aware routines such as "set up my workspace". As robots operate in shared environments, future systems must also address the challenge of managing conflicting preferences across multiple users while ensuring data privacy. We hope that VAP serves as a foundational step toward injecting user-specific signals into large VLAs, paving the way for robotic agents that can adapt to individual needs, routines, and concepts in a safe and transparent manner.

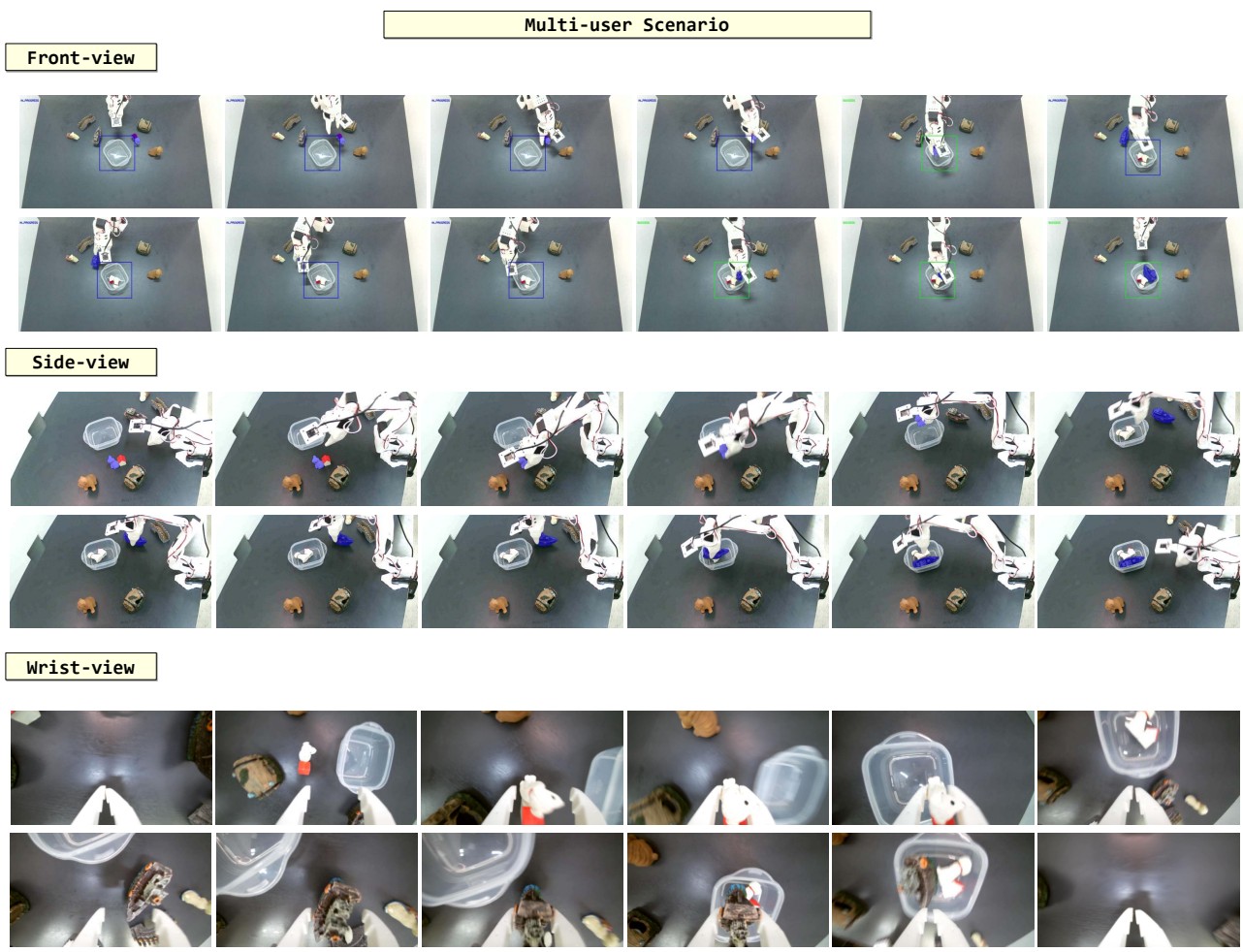

*Figure 24.* Unified sequential execution with a single mask. VAP grounds and manipulates *my brother's dog figurine*. After the first action perturbs the scene, VAP is re-run to correctly locate and mask *my ornament* for the second sub-task, illustrating robust binding under state changes.

