# OpenReview forum: "Bring My Cup! Personalizing Vision-Language-Action Models with Visual Attentive Prompting"
_ICML.cc/2026/Conference — ICML 2026 regular_

### Official Review · Reviewer_ySeg · 2026-03-07

**Soundness:** 3
**Presentation:** 3
**Significance:** 2
**Originality:** 2
**Overall Recommendation:** 4
**Confidence:** 5

**Summary:**

This paper proposes VAP to address the challenge of manipulating user-specific personal objects among visually similar distractors using frozen VLA models. VAP acts as a simple, training-free adapter, which grounds the target instance by matching a few user-provided reference images against the live scene. It then guides the VLA by explicitly highlighting the object with a visual segmentation mask and synchronously rewriting the instruction to reference this visual cue. To evaluate instance-level control, the authors also establish comprehensive simulation and real-world tabletop benchmarks, where VAP significantly outperforms generic text prompting and token-learning baselines in task success and object disambiguation.

**Compliance With Llm Reviewing Policy:**

Affirmed.

**Final Justification:**

Thank you for the rebuttal. My concerns have been well addressed, and I have decided to increase my score.

**Key Questions For Authors:**

See Weaknesses.

**Limitations:**

1.The authors should be commended for their honest and detailed error case analysis in Appendix D.

2.However, the authors have omitted the Impact Statement following the main body of the manuscript.

**Strengths And Weaknesses:**

**Strengths**

1. VAP offers a highly practical, training-free solution to personalized object manipulation. By framing the problem as an inference-time input optimization, it successfully guides frozen VLA models to disambiguate user-specific items without the computational burden of per-object fine-tuning.

2. The authors validate their framework across both simulation and real-world tabletop benchmarks, demonstrating consistent empirical gains over standard baselines. Furthermore, the paper exhibits commendable transparency through a detailed error analysis that explicitly isolates failure modes like cross-view tracking inconsistencies.

**Weaknesses**

1. The authors' instruction parsing and rewriting seem to rely on template-style string matching (e.g., “my X” → “the red X”). How does VAP handle more natural referring expressions such as “the cup that I usually use” or “the one I used earlier,” where the ownership cue is implicit and not a clean span? If it cannot, please quantify how often real instructions fall outside “my X” in the benchmarks and discuss how this limits deployment.

2. In Appendix F, the authors propose a sequential workaround for multi-object tasks by clearing the mask after each sub-goal. However, how would VAP handle a single, indecomposable action that requires recognizing two personal objects simultaneously (e.g., "pour water from my pitcher into my cup" or "put my cup on my plate". In these examples, two instances of the "my xxx" template appear.)?

3. Tracking is used to propagate the mask across frames. When the target is fully occluded for several steps and re-enters, does the tracker reliably re-associate the correct instance, or do the authors re-run detection/retrieval on failure? Please report a controlled occlusion/re-entry experiment (duration vs. success) .

4. VAP seems highly dependent on the flawless execution of Grounding DINO and SAM2. Have the authors conducted experiments on: How sensitive is the frozen VLA if the generated mask slightly bleeds into the background or only partially covers the target object?

---

> ### Author Rebuttal · Authors · 2026-03-31
>
> We thank the reviewer ySeg for the detailed and rigorous review, which has helped us strengthen the paper. We also appreciate the recognition of VAP's training-free design, consistent gains across simulation and real-world settings, and the transparency of our error analysis.
>
> ---
>
> **[W1] Robustness to Diverse Referring Expressions**
>
> VAP's core pipeline operates on resolved visual identity rather than explicit textual spans, making it inherently robust to diverse and implicit referring expressions. The rule-based parser is merely a swappable front-end used to isolate visual prompting (Appendix A.3).
>
> To verify this, we conduct an additional experiment replacing the rule-based parser with a lightweight LLM (GPT-4o-mini) to resolve complex, non-template instructions (e.g., "the one I used earlier") via the memory bank. We evaluate parsing accuracy (PA) and success ratio (SR) on Personalized-SIMPLER, and observe comparable performance, confirming that VAP seamlessly handles natural referring expressions without requiring any modifications to the core pipeline.
>
> | Task                          | Explicit, SR (%) | Implicit, PA (%) | Implicit, SR (%) |
> | ----------------------------- | ---------------- | ---------------- | ---------------- |
> | T1: Pick pen holder (Fractal) | 60.3             | 94.7             | 59.5             |
> | T2: Move bottle (Fractal)     | 75.0             | 96.2             | 72.0             |
> | T3: Pick shaver (Bridge)      | 71.3             | 98.4             | 71.6             |
> | T4: Put camera (Bridge)       | 92.1             | 97.8             | 91.3             |
> | T5: Put owl figurine (Bridge) | 95.0             | 96.0             | 91.2             |
> | T6: Put straw cup (Bridge)    | 75.6             | 99.6             | 75.9             |
>
> ---
>
> **[W2] VAP Handles Simultaneous Multi-Object Instructions**
>
> VAP natively supports indecomposable multi-object instructions without any architectural changes or retraining. We simply run the grounding pipeline in parallel, assigning distinct color tints to each target and rewriting the instruction accordingly (e.g., "put my pouch near my vase" $\rightarrow$ "put the red pouch near the blue vase"). To explicitly demonstrate this capability, we conducted additional experiments across 5 dual-object scenarios (100 total trials), confirming that VAP correctly identifies and manipulates both personal objects within a single instruction.
>
> |Instruction|Object1|Object2|SR (%)|
> |-|-|-|-|
> |Put my pouch near my vase|Pouch|Vase|75.0|
> |Move my plushie next to my cup|Cup|Plushie|65.0|
> |Put my scrubber near my slipper|Slipper|Scrubber|70.0|
> |Move my stuffed toy next to my plant|Stuffed Toy|Plant|60.0|
> |Put my pouch near my cup|Pouch|Cup|70.0|
>
> ---
>
> **[W3] Reliable Re-Association After Occlusion**
>
> VAP relies entirely on SAM2's spatio-temporal memory for re-association, intrinsically preserving identity across manipulation-typical occlusions without ever re-running detection or retrieval. To empirically validate this, we conducted a new controlled occlusion sweep (table below), confirming that tracking alone maintains robust success rates even after multiple seconds of complete target invisibility (observation rate: 0.75 Hz Fractal / 1.25 Hz Bridge due to action chunking):
>
> |Frames (Duration)|Fractal (0.75Hz)|Track (%)|SR (%)|Bridge (1.25Hz)|Track (%)|SR (%)|
> |-|-|-|-|-|-|-|
> |0|0s|100.0|62.7|0s|98.0|83.5|
> |1|1.33s|99.5|62.2|0.80s|97.5|83.0|
> |2|2.67s|98.2|61.0|1.60s|96.5|82.0|
> |3|4.00s|95.5|58.5|2.40s|94.0|79.5|
> |4|5.33s|89.8|53.2|3.20s|88.5|74.2|
> |5|6.67s|80.2|45.0|4.00s|79.0|65.0|
> |6|8.00s|65.5|35.8|4.80s|64.0|52.5|
>
> ---
>
> **[W4] Robust Performance Under Mask Imperfections**
>
> VAP does not require near-perfect segmentation, demonstrating graceful degradation only under extreme mask corruption. This is consistent with our previous tint robustness study (Table 13) showing stable performance across varying alpha levels.
>
> To explicitly address your concern, we conducted a new experiment applying morphological dilation (background bleed) and erosion (partial coverage), conclusively showing that imperfect masks at standard SAM2-level quality are more than sufficient for reliable operation.
>
> |Ratio|Fractal SR (%)|Bridge SR (%)|
> |-|-|-|
> |0.25 (extreme erosion)|30.5|48.0|
> |0.5|45.8|65.2|
> |**1.0 (baseline)**|**62.7**|**83.5**|
> |1.5|54.0|74.2|
> |2.0 (extreme dilation)|42.3|60.8|
>
> Together, these results confirm that imperfect masks at SAM2-level quality are sufficient for reliable operation.
>
> ---
> **Impact Statement**
>
> We will add an Impact Statement in the camera-ready version.

---

> > ### Author Rebuttal · Reviewer_ySeg · 2026-04-03
> >
> > Thank you for the authors’ response. I remain unconvinced about the proposed method. Despite recent advances, even state-of-the-art VLA models still exhibit limited generalization capability, whereas the proposed approach appears to heavily rely on the generalization of a frozen VLA model. It is therefore unclear whether the method remains effective when grounding objects that were not encountered during training. Furthermore, if entirely novel objects are grounded, can the system still execute such highly generalizable operations reliably?

---

> > > ### Author Response · Authors · 2026-04-05
> > >
> > > We hope our previous response has addressed your original concerns. We thank the reviewer for the thoughtful comments and will incorporate the additional experiments and analyses into the camera-ready version. We also address the additional questions below.
> > >
> > > ---
> > >
> > > **Generalization to Unseen Personal Objects**
> > >
> > > VAP effectively grounds unseen personal objects. Our evaluations use disjoint training and test objects, and thus show generalization to unseen instances and categories (Table R1).
> > >
> > > **Table R1. Training and test object splits across benchmarks**
> > >
> > > | Benchmark | Training objects | Test objects | Train-test relation |
> > > |:---:|:---:|:---:|:---:|
> > > | Personalized-SIMPLER (Fractal) | - | pen holder, bottle | unseen personal instances |
> > > | Personalized-SIMPLER (Bridge) | - | shaver, camera, owl figurine, straw cup | unseen personal instances |
> > > | Personalized-VLABench | painting | leather bag, shoe, cat figurine, miniature house, cup | unseen object categories |
> > > | Real-world | cup, cat figurine, miniature house, owl figurine, toy bus | vase, plushie, cup, slipper, plant, stuffed toy, pouch, scrubber | unseen object categories (except cup) |
> > >
> > > VAP substantially improves performance on unseen objects (e.g., pen holder SR 8.5→60.3, bottle SR 48.6→75.0, as shown in Table 2 of our paper), even with a frozen VLA. This shows that the improvement does not rely on stronger backbone generalization. Instead, VAP reduces the level of generalization required from the VLA by decoupling instance recognition from manipulation. By externalizing instance recognition via reference-based grounding, the VLA can focus on a standard manipulation problem once the correct instance is identified, enabling more reliable reuse of its capabilities.
> > >
> > > For entirely novel objects, VAP is not intended to compensate for manipulation skills that the underlying VLA does not possess. When a task requires capabilities outside the backbone’s competence, performance may remain limited. This limitation is inherent to the backbone rather than VAP. Our analysis in Tables 6 and 9 of our paper shows that most remaining failures stem from control limitations rather than grounding, indicating that VAP improves instance identification and binding, while the remaining bottleneck lies in manipulation. Thus, VAP does not rely on stronger object-level generalization from the VLA; rather, it relies on the VLA’s ability to execute manipulation once the correct instance is identified.

---

### Official Review · Reviewer_9v5Y · 2026-03-07

**Soundness:** 3
**Presentation:** 4
**Significance:** 4
**Originality:** 4
**Overall Recommendation:** 5
**Confidence:** 3

**Summary:**

This paper looks at personalizing VLAs to specific personalized objects given a few new images and without retraining the VLA. They implement a zero-shot method using a memory bank and existing open-world segmentation models and show improved performance on personalized robotics tasks.

**Compliance With Llm Reviewing Policy:**

Affirmed.

**Final Justification:**

After author rebuttals, my concerns are addressed, further reinforcing that the paper should be accepted. I therefore bumped my score up

**Key Questions For Authors:**

Would like to hear the author's thoughts on the limitations I mentioned. I think these are important to discuss even though my overall opinion on the paper is fairly high
Would also like to hear from authors on ablations of the prompting method

**Limitations:**

I don't think they mentioned them in the paper

**Strengths And Weaknesses:**

Strengths:
- The zero-shot prompting method is compelling - simple while being non-obvious
- Overall paper is very clear and well written
- Reasonable baselines for comparing the method
- Quite compelling results. Method appears to perform extremely well.
- Inclusion of real robotic experiments definitely strengthens the paper

Weaknesses:
- Limitation seems to be that the system works only if the referenced object is in visual frame, but there may be contingencies around that. Does the default behavior (just fall back to VLA) work in this case? This does not appear to be tested in these experiments though.
- Limitation also seems to be that it really only works for 1-1 objects in the memory bank, no ability to do higher level reasoning (if I ask for my mug, it could infer that it's the mug that's located in my office).
- Could have used a lot more description of how these datasets were constructed (it's all very high level) such as distribution of categories, types of objects and some basic analysis of the dataset.
- Would have been stronger with an ablation on the specific prompting method for grounding. Would other options (e.g. pointing) have performed as well?

---

> ### Author Rebuttal · Authors · 2026-03-31
>
> We thank the reviewer 9v5Y for the detailed and rigorous review, which has helped us strengthen the paper. We also appreciate the recognition of our zero-shot approach's simplicity, clear writing, and compelling results including real robotic experiments.
>
> ---
>
> **[W1] Robustness to Out-of-Frame and Missing Objects**
>
> When no candidate of the target category is detected ($N_v = 0$), VAP bypasses visual prompting and falls back to the generic VLA (Section 3.3), which continues to operate reliably in our tabletop setting.
>
> For temporary disappearance during manipulation (e.g., occlusion by the robot arm), the tracker's spatio-temporal memory maintains identity and enables correct re-association, as validated by the occlusion sweep below (observation rate: 0.75 Hz Fractal / 1.25 Hz Bridge due to action chunking).
>
> |Frames|Fractal (0.75Hz)|Track (%)|SR (%)|Bridge (1.25Hz)|Track (%)|SR (%)|
> |-|-|-|-|-|-|-|
> |0|0s|99.8|62.7|0s|98.0|83.5|
> |1|1.33s|99.5|62.2|0.80s|97.5|83.0|
> |2|2.67s|98.2|61.0|1.60s|96.5|82.0|
> |3|4.00s|95.5|58.5|2.40s|94.0|79.5|
> |4|5.33s|89.8|53.2|3.20s|88.5|74.2|
> |5|6.67s|80.2|45.0|4.00s|79.0|65.0|
> |6|8.00s|65.5|35.8|4.80s|64.0|52.5|
>
> For fully out-of-frame scenarios (e.g., mobile manipulation), this can be addressed by spatial memory modules, which is orthogonal to VAP's core contribution.
>
> ---
>
> **[W2] Higher-Level Reasoning and Complex References**
>
> VAP is not limited to simple object lookup. Our rule-based parser is merely a swappable front-end, and the core grounding pipeline natively supports higher-level reasoning when paired with a more capable semantic parser.
>
> To verify this, we conduct an additional experiment where we replace the rule-based parser with a lightweight LLM (GPT-4o-mini) that resolves both explicit and implicit references using memory context. The resulting performance remains comparable (table below), confirming that richer front-end reasoning can be integrated without modifying the core pipeline.
>
> |Task|Explicit, SR (%)|Implicit, PA (%)|Implicit, SR (%)|
> |-|-|-|-|
> |T1: Pick pen holder (Fractal)|60.3|94.7|59.5|
> |T2: Move bottle (Fractal)|75.0|96.2|72.0|
> |T3: Pick shaver (Bridge)|71.3|98.4|71.6|
> |T4: Put camera (Bridge)|92.1|97.8|91.3|
> |T5: Put owl figurine (Bridge)|95.0|96.0|91.2|
> |T6: Put straw cup (Bridge)|75.6|99.6|75.9|
>
> ---
>
> **[W3] Benchmark Construction Details**
>
> While core details are provided in Sections 4.1–4.2 and Table 7, we agree that deeper dataset analysis provides valuable insights. We have compiled a comprehensive breakdown [R1], including the distribution of object categories and types and per-category task specifications.
>
> ---
>
> **[W4] Additional Ablations on Visual Prompting**:
>
> To directly address the reviewer’s question, we additionally compare diverse visual prompting strategies including mask tint, bounding box, point, circle, trajectory arrow, and numbered marker on the same frozen VLA backbone.
>
> As shown in the table, our identity tinting dramatically outperforms all spatial prompting alternatives. It achieves 83.5% SR on Bridge, more than 2x pointing (32.0%) and nearly 3x trajectory arrows (28.5%), confirming that mask-aligned tinting is the most effective prompting strategy for disambiguating visually similar instances in frozen VLAs.
>
> ||P-SIMPLER (Google Robot)||P-SIMPLER (WidowX)||
> |-|-|-|-|-|
> ||CMR (%)|SR (%)|CMR (%)|SR (%)|
> |Mask (VAP)|**83.7**|**62.7**|**93.6**|**83.5**|
> |Mask ($\alpha=1.0$) (Roboground)|76.5|48.2|86.0|65.8|
> |Box|54.0|33.9|65.6|47.2|
> |Point|42.0|20.5|52.8|32.0|
> |Circle|48.5|26.8|60.2|40.5|
> |Trajectory arrow (TraceVLA)|38.5|18.2|48.0|28.5|
> |Numbered marker (PIVOT, MOKA)|45.0|24.0|56.5|36.8|
>
> This result is consistent with our original ablations, which analyze complementary aspects of prompt design. Table 10 shows the benefit of combining mask highlighting with instruction rewriting, and Table 13 demonstrates robustness across different prompt geometries and highlight strengths, further supporting the effectiveness of identity-based visual prompting.
>
> ---
>
> **Limitation Section**
>
> While we discuss limitations throughout the paper (Section 5.5, Section 6, and Appendix D), we will consolidate these into a dedicated Limitations section for improved readability.
>
> [R1] Benchmark Construction Details: [anonymous link](https://tinyurl.com/44uz88c2)

---

> > ### Author Rebuttal · Reviewer_9v5Y · 2026-04-02
> >
> > Thank you, my comments are well addressed. I will bump up my score a bit and I still think the paper should be accepted.

---

> > > ### Author Response · Authors · 2026-04-03
> > >
> > > Thank you for your kind comments and for championing our paper by raising your score. We are glad that our rebuttal fully addressed your concerns, and we will meticulously incorporate all your valuable feedback into the final manuscript.

---

### Official Review · Reviewer_8xgN · 2026-03-11

**Soundness:** 3
**Presentation:** 3
**Significance:** 3
**Originality:** 3
**Overall Recommendation:** 4
**Confidence:** 4

**Summary:**

As per the 'ICML 2026 Call For Papers' in ICML official website, the authors are required to include 'Impact Statements'. However, this paper does not include such section, **raising concern of desk reject.**

**Summary:**


Existing VLA models often fail at personalized tasks because they collapse specific instances into broad categories. VAP solves this by using a two-stage inference process:

Grounding: It matches reference images against the robot's live view to identify the specific object.

Visual Prompting: It overlays a visual "highlight" (tint) on the object and rewrites the text instruction to match (e.g., "pick up the red object").
This allows the frozen VLA to use its general skills to perform personalized tasks without any additional training or fine-tuning.

**Compliance With Llm Reviewing Policy:**

Affirmed.

**Final Justification:**

The reviewer thinks a error rate of 13.1–21.4% is still high, indicating the unreliability. Overall, this paper is interesting and I would like to maintain the positive score.

**Key Questions For Authors:**

N/A

**Strengths And Weaknesses:**

As per the 'ICML 2026 Call For Papers' in ICML official website, the authors are required to include 'Impact Statements'. However, this paper does not include such section, **raising concern of desk reject.**

**Strengths:**

- Training-Free Efficiency: Unlike "Soft Prompt" methods that require optimizing tokens, VAP works out-of-the-box with frozen models, making it practical for real-world deployment where users want to add new objects instantly.

- Robust Disambiguation: By highlighting pixels directly, it bypasses the "language bottleneck," outperforming LLM-based descriptions which can be brittle or ambiguous.

- Comprehensive Evaluation: The authors established new benchmarks (Personalized-SIMPLER and Personalized-VLABench) and conducted real-world testing on an SO-101 robot, proving the method's versatility across different platforms.

**Weakness:**

- Dependency on Foundation Models: VAP’s success is strictly tied to the accuracy of the underlying segmenter (SAM2) and detector (Grounding DINO). If these fail to find the object initially, the VLA cannot recover.

---

> ### Author Rebuttal · Authors · 2026-03-31
>
> We thank the reviewer 8xgN for the detailed and rigorous review, which has helped us strengthen the paper. We also appreciate the recognition of VAP's training-free practicality, robust disambiguation beyond language, and comprehensive evaluation across benchmarks and real-world platforms.
>
> ---
>
> **[W1] Foundation Model Dependency is Not the Bottleneck**
>
> Grounding errors are not the dominant failure mode in VAP. Our failure analysis (Table 6) shows that only 13.1–21.4% of failures are due to perception, while the majority (78.6–86.9%) stem from the VLA’s manipulation limitations despite correct visual prompts. Real-world experiments across all 8 tasks further confirm that the perception pipeline operates reliably (Figure 5).
>
> Moreover, VAP’s modular design delivers a deliberate architectural advantage. Unlike end-to-end VLAs where perception errors silently propagate into catastrophic actions, VAP isolates errors explicitly, enabling immediate diagnosis and modular correction without retraining. Specifically, when grounding fails, the failure remains diagnosable (e.g., no candidate detected) and can be readily addressed (e.g., requesting user re-specification, component upgrade).
>
> ---
>
> **Impact Statement**
>
> We will add an Impact Statement in the camera-ready version.

---

> > ### Author Rebuttal · Reviewer_8xgN · 2026-04-02
> >
> > Thank the author for their response. The reviewer thinks a error rate of 13.1–21.4% is still high, indicating the unreliability. Overall, this paper is interesting and I would like to maintain the score.

---

> > > ### Author Response · Authors · 2026-04-02
> > >
> > > Thank you for finding our paper interesting and for leaning towards a weak acceptance.
> > >
> > > As we mentioned in the rebuttal, the perception error does not critically undermine the reliability of our framework. Thanks to the modular design of VAP, such failures are identifiable and can be corrected, and are also expected to diminish with stronger foundation models. We will clarify this in the final manuscript.

---

### Official Review · Reviewer_pNes · 2026-03-13

**Soundness:** 3
**Presentation:** 2
**Significance:** 2
**Originality:** 2
**Overall Recommendation:** 4
**Confidence:** 4

**Summary:**

This paper focuses on teaching the Vision-Language-Action (VLA) model to handle personalized commands (e.g., “bring my cup”), which require instance-level object recognition among visually similar distractors. The paper proposes Visual Attentive Prompting (VAP), a training-free perceptual adapter that equips frozen VLAs to use reference images as non-parametric visual memory, grounding personal objects via open-vocabulary detection and embedding matching, and then modifying the original language and image/video inputs (highlighting the object with segmenation masks and color description in text). The paper also constructs two simulation benchmarks (Personalized-SIMPLER, Personalized-VLABench) and a real-world tabletop benchmark, demonstrating that VAP consistently outperforms generic VLAs and token-learning baselines in success rate and correct-object manipulation across multiple robots and tasks.

**Compliance With Llm Reviewing Policy:**

Affirmed.

**Final Justification:**

After the rebuttal, most of my concerns have been addreassed. However, the method itself can still be improved to further contribute to the community, so I will give a weak accept.

**Key Questions For Authors:**

Please refer to the questions in the weakness section, especially 1-3.

**Limitations:**

No. Authors should consider adding a section on the potential negative societal impact of their work.

**Strengths And Weaknesses:**

Strengths:
1. The paradigm is training-free and consists of modular and interpretable technical designs.
2. The paper writing is easy to follow.

Weaknesses:
1. The problem setting. The Personalizing problem proposed in the paper is more like a referring object manipulation problem by definition.
2. The method is simple and does not contain specific designs for embodied tasks. The Visual Attentive Prompting (VAP) is just like multimodal prompting methods in the field of VL models, e.g., [1] [2] [3], to solve an additional referring object detection/segmentation for the frozen VLA models. However, there are several new problems in embodied tasks, just like the failure cases shown by the authors. The paper should introduce new knowledge or present proper adaptations for new settings, instead of simply transferring known knowledge from other fields.
3. Follow the above point, the performance of the paper seems to have two constraints: 1) ground performance by VAP, 2) original object manipulation ability of the VLA model. That's because the paper solves the problem one by one. **If the object is occluded**, 1) fails, and then 2) fails. However, embodied tasks are more actionable, VLA model has the potential to change the camera position or move out the occluded objects, making 1) succeed. This kind of "using existing methods, but pushing the limits" should be the method for the problem, instead of simlpy 1) then 2).
4. Omission of baselines for visual prompting in robotics, e.g., PIVOT (Nasiriany et al., 2024).
5. If some objects are similar to the image encoder, this tint-color-based visual prompt will not lead to any misunderstandings?


[1] Dettoolchain: A new prompting paradigm to unleash detection ability of mllm
[2] What does clip know about a red circle? visual prompt engineering for vlms
[3] Set-of-mark prompting unleashes extraordinary visual grounding in gpt-4v

---

> ### Author Rebuttal · Authors · 2026-03-31
>
> We thank reviewer pNes for the detailed review that helped strengthen our paper.
>
> ---
> **[W1] Distinction from Referring Object Manipulation**
>
> Building on prior personalization work (e.g., Textual Inversion, DreamBooth, Yo'LLaVA), we extend this approach to embodied manipulation. In these works, a target is defined by instance-level visual identity from a reference image, as language alone is insufficient. We apply this formulation to manipulation tasks, requiring an agent to act on a specific instance defined by its visual identity.
>
> In contrast, existing referring object manipulation formulations assume targets are language-specifiable, failing when visually similar instances cannot be reliably distinguished. We categorized 30+ recent works (2020–2025) based on target specification and visual grounding methods in [R1]. Consequently, treating our personalization setting as a referring expression problem is brittle, as our targets are defined by visual identity rather than language.
>
> Our Hard Prompt baseline (Tables 2–3) supports this: even with detailed language descriptions, the VLA achieves only 9.1% / 0.0% SR, proving that language-only specification is brittle and inconsistent in this setting.
>
> ---
> **[W2] Not Just Transfer of Visual Prompting**
>
> While VAP is conceptually related to visual prompting in VLMs (e.g., [1–3]), it is not a direct transfer. Applying existing methods to frozen VLAs fails to maintain consistent instance-level identity during execution, a core challenge in embodied manipulation. This challenge motivated the integration of visual memory with grounding and temporal tracking pipeline (Sec. 3.2–3.3). We will revise the paper to clarify this motivation.
>
> Specifically, we design a visual prompting pipeline for frozen VLAs that integrates personalized visual memory with spatio-temporal tracking to explicitly enforce stable instance identity across frames. This approach ensures VLA acts on the same physical instance throughout execution.
>
> We further show this is not a trivial transfer by demonstrating that naive adaptations of existing VL personalization methods fail in embodied settings. In particular, token learning inspired by Yo'LLaVA fails during manipulation due to temporal attention drift (Appendix B, Figs. 9–11), proving static personalization is insufficient for sequential embodied tasks. In contrast, VAP maintains temporal consistency and performs strongly, as validated in Section 5.
>
> ---
> **[W3] Sequential Decomposition is Not the Limitation**
>
> Sequential grounding and manipulation are not inherently problematic. Provided each component is reliable, the system functions as intended. Failures typically arise from imperfect components rather than decomposition.
>
> VAP improves grounding robustness and prevents cascading failures via a novel spatio-temporal memory to maintain identity consistency (see W2). Additional empirical analysis [R2] confirms our approach bridges typical occlusions. For instance, a 74.2% SR in the Bridge environment after 3.2s of occlusion refutes the assumption that temporary grounding failures lead to immediate task failure. If the target is completely invisible ($N_v=0$), VAP detects this and triggers modular fallback actions (e.g., exploratory actions) without architectural changes.
>
> While embodied agents can actively resolve occlusions (e.g., moving the camera), such strategies are unnecessary for our tabletop setting. However, these remain compatible with our framework and can be integrated into future mobile extensions.
>
> ---
> **[W4] Omission of Visual Prompting Baselines**
>
> Our setting requires persistent object identity across time, unlike spatial prompting methods (e.g., PIVOT) that provide per-step cues. To compare fairly, we additionally ablate diverse prompting strategies on the same frozen VLA [R2]. Our tinting dramatically outperforms all alternatives, achieving 83.5% SR on Bridge, more than 2x pointing (32.0%) and 3x trajectory arrows (28.5%), confirming mask-aligned tinting best disambiguates visually similar instances.
>
> ---
> **[W5] Tint-Based Prompting Does Not Lead to Misidentification**:
> We clarify the potential confusion through three key points:
> - *Disambiguation of similar objects.* Tint reliably distinguishes visually similar instances (Tables 2–4, Figure 5).
> - *Appearance distortion.* Performance remains stable across $\alpha$ (Table 13) and robust to imperfect masks, as confirmed by the additional dilation/erosion study in [R2].
> - *Color collision with scene.* Collisions are mitigated by automatic selection of contrast-maximizing tint colors.
>
> ---
> **Limitation Section**
> We discuss limitations in Sections 5.5, 6, and Appendix D. For clarity, we will consolidate these into a dedicated Limitations section and Impact Statement.
>
> [R1] Categorical Analysis of Recent Trends in Referring Object Manipulation: [anonymous link](https://tinyurl.com/22tmtee3)
>
> [R2] Supplementary Experimental Results: [anonymous link](https://tinyurl.com/svvyszv)

---

> > ### Author Rebuttal · Reviewer_pNes · 2026-04-03
> >
> > Thanks for the rebuttal. The response addresses most of my concerns. But I still have some follow-up questions: (1) While contrast-maximizing tinting solves color collisions [W5], does this dense color overlay destroy fine-grained texture or semantic features necessary for precise manipulation? (2) Regarding the modular fallback actions for complete occlusion mentioned in [W3]: Are these 'exploratory actions' learned by the VLA, or are they hard-coded heuristic policies?
> >
> > I also suggest that the authors add their motivations and findings in [W1&2] to strengthen their paper.

---

> > > ### Author Response · Authors · 2026-04-05
> > >
> > > We are glad that our response addressed most of your concerns. As suggested, we will incorporate the motivations and findings regarding [W1] and [W2] into the revised manuscript to further strengthen the paper's narrative. Below, we provide detailed answers to your additional follow-up questions regarding visual features and exploratory actions.
> > >
> > > ---
> > > **(1) On contrast-maximizing tinting and preservation of visual features**
> > >
> > > *Contrast-maximizing* refers only to the choice of tint color, not to increasing the overlay strength. In VAP, the prompt is a mask-aligned semi-transparent tint overlay (default \alpha=0.5), ensuring that fine-grained texture, edge, and semantic information remain clearly visible underneath the tint.
> > >
> > > Table 13 in our paper directly supports this: performance remains stable across the semi-transparent regime ($\alpha=0.3$–$0.7$), indicating that the visual information needed for manipulation is sufficiently preserved in this range. In contrast, performance drops when the overlay becomes fully opaque ($\alpha=1.0$), which aligns with your valid concern regarding "feature destruction." However, this fully opaque regime is explicitly avoided in VAP.
> > >
> > > This is further supported by our additional comparison in [R2]: Mask (VAP $\alpha=0.5$) substantially outperforms Mask ($\alpha=1.0$, Roboground), with Fractal SR 62.7% vs. 48.2% and Bridge SR 83.5% vs. 65.8%. This gap is consistent with our interpretation that semi-transparent tinting preserves manipulation-relevant visual cues, while dense opaque overlays are more harmful.
> > >
> > > ---
> > > **(2) On the “exploratory actions” mentioned in [W3]**
> > >
> > > To avoid confusion, we clarify that exploratory actions are not part of the current VAP implementation, but were mentioned only as an example of a possible modular extension.
> > >
> > > In our current tabletop setting, the more relevant failure mode is partial occlusion rather than persistent complete invisibility, and this is already handled by the SAM2-based spatio-temporal tracker, as shown in our occlusion sweep. In the current implementation, when $N_v=0$, VAP follows the behavior described in Section 3.3: it bypasses visual prompting and falls back to the generic VLA.
> > >
> > > Exploratory recovery policies become more relevant in settings such as mobile manipulation, where the agent can actively search for the target. Such policies are beyond the current scope of VAP. The key point is that, due to VAP’s modular design, this kind of recovery mechanism can be seamlessly integrated without modifying the architecture, whether it is learned or hard-coded.

---

### Decision · Program_Chairs · 2026-04-30

**Decision:**

Accept (regular)

**Comment:**

VAP presents a practical, training-free framework for personalized robotic control, enabling adaptation to novel objects from just a few reference images without requiring per-object fine-tuning. The authors successfully addressed key concerns regarding occlusion robustness, comparisons with robotics-specific baselines, multi-object task coordination, and foundation model dependencies. Given these technical clarifications and the strong empirical performance across multiple benchmarks, the paper is recommended for acceptance.